# *hoxc12/c13* as key regulators for rebooting the developmental program in *Xenopus* limb regeneration

Aiko Kawasumi-Kita[1,6], Sang-Woo Lee[1,6], Daisuke Ohtsuka[1], Kaori Niimi[1], Yoshifumi Asakura[1], Keiichi Kitajima[1,2], Yuto Sakane[3], Koji Tamura[2], Haruki Ochi[4], Ken-ichi T. Suzuki [3,5] & Yoshihiro Morishita [1,6] ✉

During organ regeneration, after the initial responses to injury, gene expression patterns similar to those in normal development are reestablished during subsequent morphogenesis phases. This supports the idea that regeneration recapitulates development and predicts the existence of genes that reboot the developmental program after the initial responses. However, such rebooting mechanisms are largely unknown. Here, we explore core rebooting factors that operate during *Xenopus* limb regeneration. Transcriptomic analysis of larval limb blastema reveals that *hoxc12/c13* show the highest regeneration specificity in expression. Knocking out each of them through genome editing inhibits cell proliferation and expression of a group of genes that are essential for development, resulting in autopod regeneration failure, while limb development and initial blastema formation are not affected. Furthermore, the induction of *hoxc12/c13* expression partially restores froglet regenerative capacity which is normally very limited compared to larval regeneration. Thus, we demonstrate the existence of genes that have a profound impact alone on rebooting of the developmental program in a regeneration-specific manner.

Amphibians have long been studied as model animals in regeneration research because of their prominent ability to regenerate various organs. In particular, the limbs of salamanders, such as axolotls and newts, are an intensively studied system (see Stocum[1] for review). Amputation of their limbs triggers initial regenerative responses: wound healing, generation and aggregation of blastema cells, and formation of the apical ectodermal cap (AEC). Once the initial blastema structure is formed, morphogenesis with growth and patterning begins. To date, hundreds of studies have cumulatively identified the genes and signaling pathways responsible for the regeneration process. These include nerve, metabolic, and immune factors especially important for earlier phases of regeneration[1]. In addition, secreted molecules such as *shh* and *fgf*s, as well as transcription factors, including *hox* family essential for normal development, have also been shown to play important roles in the morphogenetic phases of regeneration. Recent single-cell transcriptomic analysis has shown the genome-wide similarities in gene expression patterns between the morphogenesis phase of regeneration and development[2–4], supporting the idea that regeneration, after an initial response to injury, recapitulates development at the molecular level.

Anuran amphibians, such as *Xenopus*, have also been widely studied due to their life-stage dependent regenerative abilities[5]. Larvae

[1]Laboratory for Developmental Morphogeometry, RIKEN Center for Biosystems Dynamics Research, Kobe 650-0047, Japan. [2]Department of Ecological Developmental Adaptability Life Sciences, Graduate School of Life Sciences, Tohoku University, Sendai 980-8578, Japan. [3]Graduate School of Science, Hiroshima University, Higashihiroshima, Hiroshima 739-8526, Japan. [4]Institute for Promotion of Medical Science Research, Faculty of Medicine, Yamagata University, 2-2-2 Iida-Nishi, Yamagata 990-9585, Japan. [5]Emerging Model Organisms Facility, Trans-scale Biology Center, National Institute for Basic Biology, National Institutes of Natural Sciences, Okazaki, Aichi 444-8585, Japan. [6]These authors contributed equally: Aiko Kawasumi-Kita, Sang-Woo Lee, Yoshihiro Morishita. ✉e-mail: yoshihiro.morishita@riken.jp

possess a greater regenerative ability, whereas when a limb is amputated in adults or juvenile frogs (froglets) after metamorphosis, early responses to injury and blastema formation occur but proper morphogenesis fails to proceed thereafter, resulting in a rod-like structure consisting of cartilage and skin referred to as a spike (Fig. 1A). For this reason, the *Xenopus* froglet/adult limb has been widely studied as a

potential gain-of-function model for strategies that may enhance regenerative abilities, with a view to future application in mammals[6–10]. Tissue engineering approaches have been successful in improving limb regeneration[11–14]. For example, by transplanting a mixture of isolated multipotent larval blastema cells and a cocktail of multiple secreted factors to the stump of amputated adult *Xenopus* limbs,

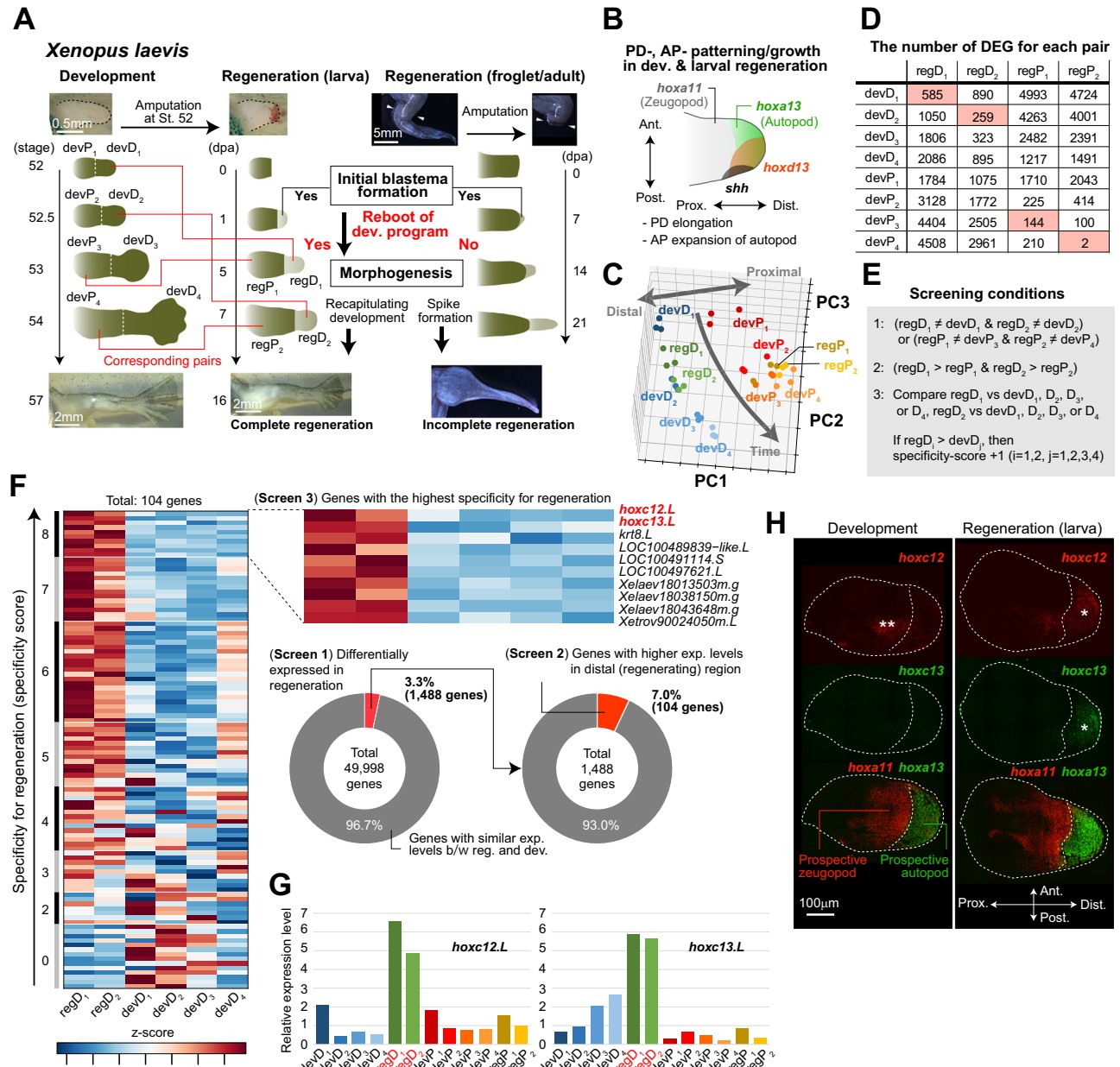

**Fig. 1 | Comparative transcriptomic analysis revealed that *hoxc12* and *hoxc13* expression show the highest regeneration specificity. A** *Xenopus* limb development and regeneration. From an amputated larval limb bud, a complete limb regenerates, whereas from an amputated adult/froglet limb, an initial blastema is formed but subsequent morphogenesis is incomplete and terminates at spike formation. Red lines indicate the corresponding stage pairs between development and regeneration based on similarities in gene expression patterns shown in (**D**). **B** Axial patterning and growth observed in normal limb development and larval regeneration. Appropriate *hox* genes and morphogen expression are essential for normal morphogenesis including proximal-distal (P-D) elongation and anterior-posterior (A-P) autopod expansion. **C** Principal component analysis. **D** Number of genes with different expression levels for each pair of samples (FDR < 0.01). Since the number itself depends on the FDR-value as a threshold, the order of the numbers is of interest here. Pink shading indicates the development sample with

the closest gene expression pattern to each regeneration sample. **E** Screening conditions for the three steps adopted. **F** The three screening steps for the detection of genes showing regeneration-specific expression changes. See the text and "Methods" section for details. The color in the leftmost panel shows the z-score for gene expression levels among the 6 samples. **G** Relative expression levels of *hoxc12.L* and *hoxc13.L* during *Xenopus* limb development and regeneration (the means of triplicate data). **H** Spatial patterns of *hoxc12.L*, *c13.L*, *a13.L* (an autopod marker), and *a11.L* (a zeugopod marker) expression within the limb bud during development and in the larval regenerating blastema. Both *hoxc12* and *hoxc13* are expressed in the prospective autopod region in the regenerating blastema (*), while during development, only *hoxc12* shows clear expression restricted to the zeugopod (**). Each experiment was independently repeated three times for each gene with similar results. Source data are provided as a Source Data file.

cartilage-to-bone differentiation and finger-like structures with joints can be formed[11]. In addition, artificial hyperinnervation[12] or the application of a hormone via a wearable bioreactor can induce branching of distal cartilage[13,14]. These studies demonstrate that it is possible to reactivate the developmental program (to some extent) even in adult tissues, given appropriate input signals.

To enhance the regenerative capacity in *Xenopus* froglets/adults, a more ideal approach might involve a detailed comparative analysis of the regenerative processes between urodele amphibians, which can regenerate limbs throughout their life cycle, and *Xenopus* adults. For instance, Lin et al.[4] conducted single-cell transcriptome analysis on tissues during the adult limb regeneration processes of *Xenopus* and the axolotl, and revealed that dedifferentiation of the connective tissues in *Xenopus*, which is the main constituent of blastema, is incomplete[4]. If molecular manipulation could achieve complete dedifferentiation in the *Xenopus* adult blastema, it might enable complete limb regeneration through the self-organization of dedifferentiated multipotent cell populations.

Another ideal approach would involve evoking intrinsic regenerative mechanisms in froglets/adults that normally function during the *Xenopus* larval stage when regenerative ability is much higher. In this study, our goal was to investigate this possibility. We began by unraveling the molecular mechanisms governing how developmental processes might be intrinsically rebooted in normal larval regeneration, a process that had not been previously elucidated. If such rebooter genes exist, their function and timing could play out in multiple ways. For instance, they may directly regulate dedifferentiation during the very early phase of regeneration, just after injury. Alternatively, they may reactivate developmental programs and promote morphogenesis, including axial patterning and growth dynamics similar to those observed during development (Fig. 1B), after dedifferentiation and initial blastema formation. The latter case may involve molecules that promote redifferentiation. Compared to dedifferentiation processes, there is little information regarding the self-organization processes of limb morphology during regeneration. Additionally, in *Xenopus* froglet/adult limb regeneration, although dedifferentiation seems to be incomplete, both wound healing and early blastema formation still occur. These provided us with sufficient motivation to explore the key regulators that function during morphogenesis after an initial response to injury. Specifically, we sought to perform a comparative transcriptomic analysis between the limb development and the morphogenesis phases of larval limb regeneration, rather than the very early phase of blastema formation. The presumptive rebooter genes may function similarly during both limb development and regeneration. However, in this study, we sought to explore the existence of regeneration-specific factors, distinct from gene sets known to have significant impacts on normal limb development, e.g., *shh* and *fgf*s. Previous omics-studies on amphibian limb regeneration using microarray and/or RNA-seq succeeded in differential expression analysis with classification/clustering into gene groups based on gene ontology and expression levels[2–4,15–22], but no genes that satisfy these conditions have been found to date.

Our bioinformatic analysis showed that *hoxc12* and *hoxc13* expression have the highest regenerative specificity. Subsequent loss-of-function analysis by knocking out each gene using a genome editing technique showed that *hoxc12/c13* expression was critical for reactivating tissue growth. This was especially true in the prospective autopod, and in reestablishing the expression patterns of genes involved in the regulatory networks that function for axial patterning during normal limb development. In addition, we showed that *hoxc12/c13* expression had no effect on either developmental processes or early events of regeneration (i.e., initial blastema formation). Finally, gain-of-function analysis, where we induced the expression of each gene in transgenic animals, showed that *hoxc12/c13* can improve regenerative capacity after metamorphosis. This included a shift in the

gene expression state to that of the developmental limb bud, distal branching of cartilage, and enhanced nerve formation. Thus, we demonstrated the existence of key regulators for rebooting the developmental program that function both intrinsically and in a regeneration-specific manner.

## Results

### Regeneration-specific expression of *hoxc12/c13*

Against the above background, we began with transcriptomic analysis to assess the regenerative specificity of the expression of each gene in *Xenopus laevis*. From developing limbs, cDNA samples of distal (including the prospective autopod and zeugopod regions) and proximal (stylopod) tissues at 4 different time points (St. 52, 52.5, 53, and 54) were collected for analysis (i.e., 8 different samples representing different stages/regions of development); in Fig. 1A, these samples are termed $devD_i$ or $devP_i$, where $i = 1, 2, 3$ or 4 indicating their temporal order. For regenerating limbs, larval blastema of limb buds amputated at St. 52 were used from which complete limb regeneration is achieved. Specifically, distal (i.e., the regenerating region) and proximal (stump) cDNA samples were collected at two different time points in which the size and shape of the regenerating blastema was similar to that of the distal limb bud at St. 52 or 52.5 of development (i.e., 4 different samples representing different stages/regions of regeneration); in Fig. 1A, these samples are termed $regD_i$ or $regP_i$, where $i = 1, 2$. The shape and size of the $regP_1$ and $regP_2$ tissues were very close to those of $devP_3$ and $devP_4$, respectively (Fig. 1A). In total, 12 varying cDNA samples were prepared, and sequencing was performed in triplicate (see "Methods" section).

First, we conducted a principal component analysis and confirmed that each triplicate dataset was located in close proximity within the principal component space, ensuring reproducibility (Fig. 1C). Additionally, in the principal component space, developmental data were neatly arranged along the time axis and proximal-distal (P-D) axis (Fig. 1C). As expected, each regeneration sample ($regD_1$, $regD_2$, $regP_1$, or $regP_2$) showed a similar expression pattern to that of the developmental sample with similar tissue shape and size (i.e., the pairs $regD_1/devD_1$, $regD_2/devD_2$, $regP_1/devP_3$, and $regP_2/devP_4$, which are referred to as corresponding pairs for convenience) (Fig. 1A and Supplementary Fig. 1). This similarity was further validated by quantifying the number of differentially expressed genes across all pairs between development and regeneration samples (Fig. 1D).

Next, to identify genes showing regeneration-specific expression changes, we performed three kinds of screening. As a first screen, we obtained 1488 genes differentially expressed during development and regeneration by comparing the expression level of each gene within each corresponding pair (Fig. 1E, F, and "Methods" section). While there were several possible options for subsequent screening, here, as the second screen we chose 104 genes that showed higher expression levels in the distal region than proximal region during regeneration (Fig. 1E, F). Lastly, we assigned a score for regeneration specificity to each gene selected in the second screen in the following manner: comparing each sample from a regenerating region (i.e., $regD_1$ or $regD_2$) with all four samples from the distal region of the developing limb bud ($devD_i$, $i = 1, 2, 3, 4$), one point was assigned to each gene when it showed higher expression during regeneration such that each gene could have a maximum of 8 points (Fig. 1E, F; "Methods" section; see also Supplementary Fig. 2 and Supplementary Data 1 for the name and ontology of 104 genes, respectively).

Consequently, we identified 10 genes with the highest regeneration specificity scores (Fig. 1F), two of which were transcription factors, *hoxc12.L* and *hoxc13.L* (abbreviated as *hoxc12* and *hoxc13* below). Comparing the relative expression levels of *hoxc12* and *hoxc13* (averaged across triplicates) from all 12 sample types, those from regenerating tissues ($regD_1$ and $regD_2$) were several to ten times higher than

the others (Fig. 1G). In addition, RNAscope assays, performed to visualize their spatial patterns of expression, showed that both *hoxc12* and *hoxc13* are expressed in the prospective autopod region in the regenerating blastema, while during development, only *hoxc12* shows clear expression restricted to the zeugopod (Fig. 1H and Supplementary Fig. 3). Thus, we chose these two genes as targets for the following functional analysis. This choice was also supported by previous reports that the expression of *hoxc13* is upregulated during limb regeneration in salamanders[23] and that inhibiting *hoxc13* using morpholinos repressed tissue growth during fish fin regeneration[24].

## Regeneration-specific function of *hoxc12/c13*

To examine the functions of the *hoxc12* and *hoxc13* genes in both limb development and larval regeneration, we knocked out each of them using the CRISPR/Cas9 system (Fig. 2A); the diploid *X. tropicalis* was used here in place of the pseudo-tetraploid *X. laevis*. The ATG region of each gene was chosen as a target site (Fig. 2A, and "Methods" section). The left hindlimb of each *hoxc12/c13* mutant F1/F2 tadpole was amputated at the prospective knee level around St. 52, and its regeneration process was observed until the regenerating area achieved a morphology equivalent to that during St. 55 to 59 in normal

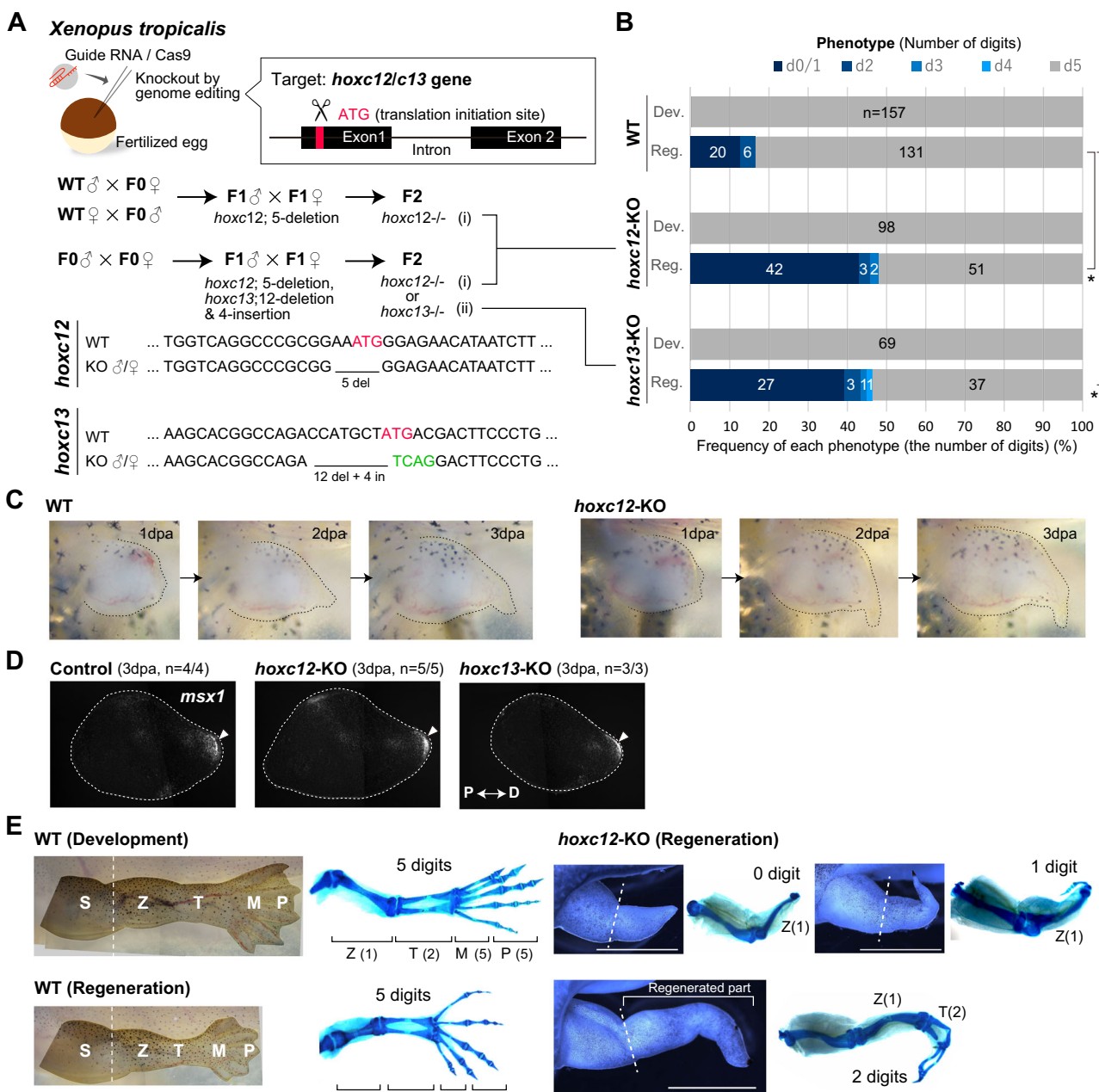

**Fig. 2 | Knockout of *hoxc12/c13* by genome editing and its effects on larval limb regeneration. A** Knockout of *hoxc12* and *hoxc13* genes by genome editing. ATG of each gene was chosen as the target site, and the diploid *Xenopus tropicalis* was used. **B** Effects on phenotypes. Knocking out *hoxc12/c13* had no effect on limb development but did affect larval regeneration. Phenotypes were evaluated based on the number of digits, and digit formation was determined by the existence of a nail at its tip. Statistical test (binomial test, two-sided) *$p = 5.7 \times 10^{-13}$, **$p = 7.6 \times 10^{-9}$. **C** Knocking out *hoxc12/c13* had no effects on initial blastema formation, at least morphologically. **D** The expression pattern of *msx1*, a typical marker gene for the initial phase of regeneration, is normal even in the *hoxc12*KO/*c13*KO individuals. Control: *hoxc12* and *hoxc13* double heterozygous individuals (*hoxc12^{+/−} hoxc13^{+/−}*). **E** Examples of limb morphologies and cartilage patterns in normal development (left upper), normal larval regeneration (left lower), and in regeneration of *hoxc12*KO individuals (right; see Supplementary Fig. 4B for phenotypes of *hoxc13*KO individuals). *hoxc12*KO: *hoxc12^{−/−}*; *hoxc13*KO: *hoxc13^{−/−}*.

development (i.e., until 10–21 days post-amputation [dpa]), while the right hindlimb of the same individual was used as a control to determine the effect on developmental processes.

Results showed that knocking out either gene had no influence on developmental processes (*hoxc12* KO: *n* = 0/98, *hoxc13KO*: *n* = 0/69, and WT: *n* = 0/157; Fig. 2B and Supplementary Fig. 4A), which is consistent with a previous report showing no significant impact on limb development in *HoxC*-cluster knockout mice[25]. In addition, the *hoxc12/c13* knockout also appeared to have no influence on initial blastema formation. Morphologically, some characteristic events occurring in the first few days after amputation, such as wound healing and cone-shaped blastema formation with thickening of the distal-most epithelium, were still observed (Fig. 2C) and a typical marker gene for the initial phase of regeneration, *msx1*[26], was expressed normally (*hoxc12* KO: *n* = 5/5, *hoxc13KO*: *n* = 3/3, and WT: *n* = 4/4; Fig. 2D). Four days after amputation, the morphological differences from normal regeneration become clear, and severe defects, especially in autopod formation, were observed in the knockout tadpoles (Fig. 2E and Supplementary Figs. 4–6); *hoxc12* KO: *n* = 47/98, *hoxc13*: *n* = 32/69, and WT: *n* = 26/157; Fig. 2B shows the frequency of defects characterized by the number of digits, i.e., 0–5. The frequency of defects slightly differed depending on the position and amount of the sequence deleted/inserted by genome editing, but, in general, approximately 40–50% of tadpoles (2.5–3 times more than WT) showed abnormalities (Supplementary Fig. 7). Moreover, the patterns and frequencies of abnormalities were similar in *hoxc12*KO and *hoxc13*KO individuals (Fig. 2B, E, and Supplementary Fig. 4B), suggesting that both genes have similar functions during limb regeneration.

In this manner, *hoxc12* or *hoxc13*, which are expressed within the prospective autopod region in a regeneration-specific manner, are not required for limb development and initial blastema formation, but are important for the morphogenesis phase of regeneration.

## Role of *hoxc12*/*hoxc13* in reactivating patterning and growth

To investigate the contribution of *hoxc12*/*hoxc13* to the regulation of other genes during larval limb regeneration, we next performed a comparative analysis of the expression patterns of genes that are essential for axial patterning and morphogenesis during development (specifically, *shh*, *hoxd13*, *hoxa13*, *hoxa11*, and *fgf8*) between regenerating limb buds of control and *hoxc12*KO/*hoxc13*KO individuals. First, RNAscope assays were performed to determine their spatial patterns of expression (Fig. 3A, Supplementary Fig. 8, and "Methods" section). The expression pattern of each gene was classified into three categories (normal, mild, and severe) based on the intensity of expression and the degree of reduction in expression range. The morphological phenotypes of the regenerating limb buds were also classified into three categories (normal, mild, and severe) based on size and shape. For *shh*, *hoxd13*, *hoxa13*, and *fgf8*, which function in the distal and/or posterior region during normal development, the frequency of individuals showing mild and severe patterns was clearly higher in the *hoxc12*KO/*hoxc13*KO population (Fig. 3B). The frequency of expression abnormalities was about 40–50%, which was consistent with the frequency of morphological abnormalities based on the number of digits (Fig. 2B). In particular, *shh* and *hoxd13*, which are expressed in a posteriorly biased manner during normal development, showed almost no expression in any of the individuals (Fig. 3B, *hoxc12*KO: *n* = 6/6, *hoxc13*KO: *n* = 7/7) that showed severe morphological phenotypes in the *hoxc12*KO/*c13*KO population, demonstrating that *hoxc12*/*c13* plays an important role in re-patterning along the A-P axis during larval limb regeneration.

Regarding the P-D patterning, no individuals showed a reduction in the amount or area of *hoxa11* expression (thus classified as normal), but importantly, *hoxa11* was expressed up to the distal-most region in all individuals with severe morphological phenotypes although it should typically be restricted to the prospective zeugopod, as

observed in normal development and regeneration, for proper P-D anatomical regionalization (Fig. 3B). In some individuals with severe morphological phenotypes, *hoxa13*, an autopod marker, was expressed in the distal-most region, but its expression was weak and/or overlapped with that of *hoxa11* (Supplementary Fig. 8). During normal development, *hoxa11* is expressed in the distal half (including the distal-most region) of the limb bud during early development, and later, when *hoxa13* expression begins in the distal-most region, exclusive expression of *hoxa13* in the prospective autopod and *hoxa11* in the zeugopod are achieved (Fig. 3A). The importance of the mutually exclusive expression of *hoxa11* and *hoxa13* during limb development has also been studied in the context of tetrapod evolution[27]. Thus, in *hoxc12*KO/*13*KO individuals, the failure of P-D regionalization leads to the severe autopod phenotype.

Next, we performed bulk-transcriptome analysis to quantitatively compare gene expression profiles between control and *hoxc13*KO tadpoles with normal or severe morphological phenotypes (Fig. 3C, D). Principal component analysis revealed that the expression profile of KO individuals with a normal phenotype closely resembled that of controls, while the profile of KO individuals with a severe phenotype distinctly differed from both (Fig. 3C). In particular, when examining the relative expression levels of the gene group related to axial patterning among all samples, which has been well-investigated in limb development research, all genes except for proximal factors (*meis1/2*) showed a clear decrease in KO individuals with severe phenotypes (Fig. 3D). This demonstrates that *hoxc13* controls the process of rebooting for a set of genes involved in morphogenesis, rather than activating a specific signaling pathway. We also conducted qPCR assays for *hoxc12*KO individuals (Supplementary Fig. 9), and the effects on the expression levels of typical patterning genes (*shh*, *hoxd13*, *hoxa13*, and *fgf8*) were in good agreement with the results of bulk-transcriptome analysis for *hoxc13*KO individuals. In particular, *shh* and *hoxd13* showed binary (i.e., all or none) expression, and the expression of both genes was almost undetectable in many individuals (*n* = 13/18), as seen in the RNAscope assay (Supplementary Fig. 9).

In the *hoxc12*KO or *hoxc13*KO population, the frequency of individuals with significantly smaller-sized blastema increased. The above gene expression analysis showed a reduction in the expression of patterning genes associated with the regulation of cell proliferation, such as *shh* and *fgf8*, therefore we counted PH3-positive cells in the prospective autopod region as defined by the presence of *hoxa13* expression (Fig. 3E). Results showed that the number of proliferating cells in the *hoxa13*-expressing region was significantly reduced, while in the other proximal tissues no clear difference was observed. This result indicates that *hoxc12/c13* is involved in regulating proliferation to ensure a sufficiently sized autopod is formed to enable normal digit patterning by reactivating the distal-posterior patterning genes. Taken together, these results show that *hoxc12/c13* expression is essential for the reestablishment of A-P and P-D patterning in blastema and autopod growth during larval limb regeneration (Fig. 3F).

## *hoxc12*/*c13* induction improves froglet regenerative abilities

As stated earlier, the regenerative abilities of *Xenopus* are significantly reduced after metamorphosis (Figs. 1A and 4A). On the basis of the above results from transcriptome and loss-of-function analyses, we next examined how *hoxc12/c13* over-expression in amputated froglet limbs affects the regeneration process. For this purpose, we created transgenic (Tg) *X. laevis* with heat-shock-inducible (heat-shock-protein 70 (hsp70) promoter) *hoxc12/c13* linked to a reporter GFP via the 2A peptide ("Methods" section). After amputation at the middle of the hindlimb zeugopod of F0 and F1 transgenic froglets, we applied localized heat shock to the blastema by soaking the tissue in agarose gel at 34–37 °C (Fig. 4B, Supplementary Fig. 10, and "Methods" section). We also evaluated the effect of varying the number and duration of the heat shocks (Supplementary Fig. 10, and "Methods" section). In

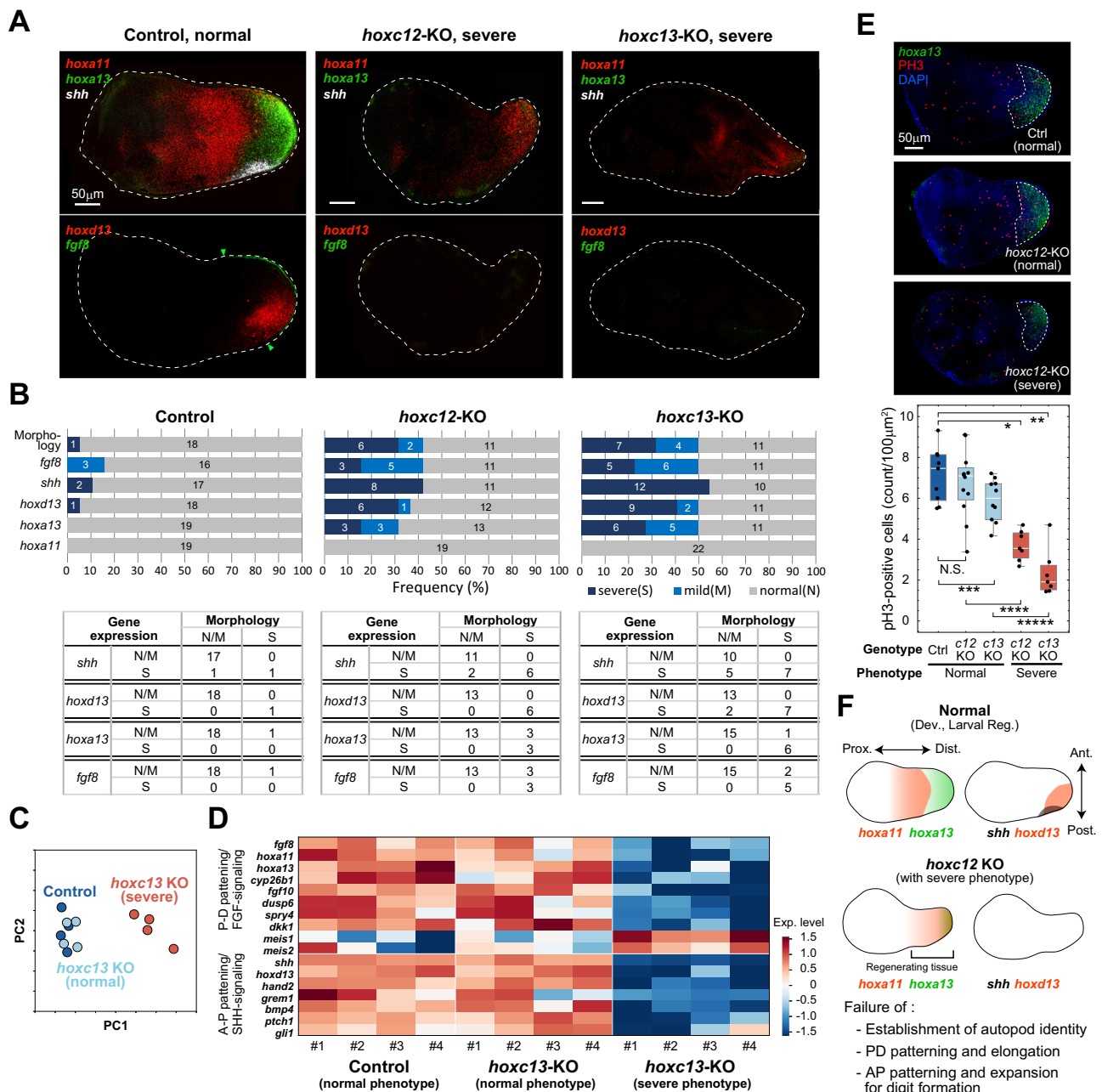

**Fig. 3 | Roles of *hoxc12*/*c13* in reactivating patterning/growth during larval limb regeneration. A** Typical spatial expression of patterning genes in the blastema of control (*hoxc12*⁺/⁺), *hoxc12*KO (*hoxc12*⁻/⁻), and *hoxc13*KO (*hoxc13*⁻/⁻) individuals at 4 dpa. **B** Statistics for the phenotypes in blastema morphology and gene expression for control (*hoxc12*⁺/⁺), *hoxc12*KO (*hoxc12*⁻/⁻), and *hoxc13*KO (*hoxc13*⁻/⁻) individuals. Biologically independent samples were used. N: normal; M: mild; S: severe. **C** Principal component analysis. **D** Comparative analysis of transcriptome data for typical patterning genes that function during limb development. **E** Effects of

*hoxc12*/*c13* knockout on cell proliferation in the prospective autopod region. Ctrl: *hoxc12*⁺/⁻. Box-plot elements: center line, median; box limits, upper and lower quartiles; whiskers, max/min. Statistical test (Student's *t*-test, two-sided): *$p = 2.44 \times 10^{-5}$; **$p = 2.43 \times 10^{-6}$; ***$p = 0.037$; ****$p = 2.81 \times 10^{-4}$, *****$p = 7.64 \times 10^{-6}$. All data were obtained from biologically independent samples: $n = 9$ (Ctrl); $n = 12$ (c12KO, normal); $n = 10$ (c13KO, normal); $n = 7$ (c12KO, severe); $n = 7$ (c13KO, severe). **F** Summary of phenotypes. Source data are provided as a Source Data file.

all cases, GFP expression was confirmed under a stereo microscope within 12 h after the first heat shock. Interestingly, after early blastema formation, a bulge, which was clearly wider in the AP direction than in wild-type spikes, was observed at the distal blastema. With continuing heat-shock induction of *hoxc12*/*c13* (e.g., for several weeks), the AP-expanded bulge was maintained, while its elongation in the PD direction was suppressed (Fig. 4B). When the heat-shock induction was stopped, the blastema began to elongate again, but no clear branched or segmented cartilage was observed at the spike tip (Fig. 4B). This result was qualitatively the same for different trials in which the

number or timing of the heat-shock inductions was varied (Supplementary Fig. 10).

Unexpectedly, however, in one-third of the hsp-*hoxc12*/*c13* Tg froglets without artificial heat-shock induction, trifurcation or bifurcation of the cartilage occurred at the distal tip of the blastema (*hoxc12*Tg: $n = 10/30$, *hoxc13*Tg: $n = 7/21$; Fig. 4C, D, Supplementary Fig. 11, Supplementary Movie 1). In addition to the branched cartilage, the distal blastema of some of Tg individuals showed a paddle-like shape with a significantly wider AP width, which was not observed in normal spikes. Consistent with this finding, the cell proliferation rate in the blastema of

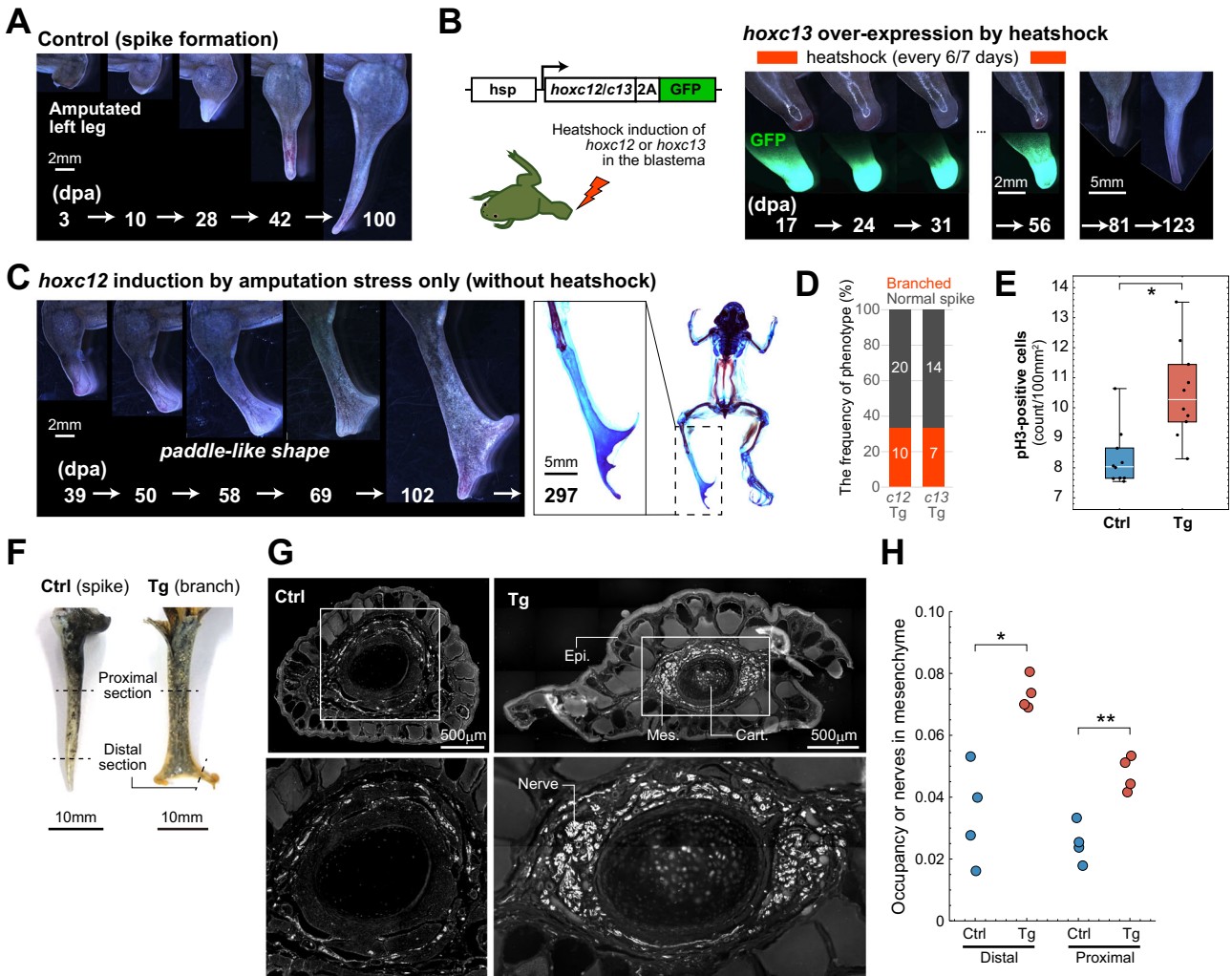

**Fig. 4 | The induction of *hoxc12/c13* expression improves regenerative abilities after metamorphosis. A** Morphological changes during spike formation from an amputated normal limb; dpa: days post-amputation. **B** Construct used to generate transgenic *Xenopus laevis* (left) and morphological changes in a blastema with *hoxc13* over-expression by heat shock (right). **C** Morphological changes in a blastema under *hoxc12* induction through amputation stress only (i.e., without heat shock; left). The shape of the blastema was paddle-like with branched distal tips, clearly different from that of normal spikes. Alcian blue/alizarin red staining of cartilage and bone (right). The branched distal cartilage tips had no clear segments and did not show bone differentiation. **D** The frequency of individuals showing branching of distal cartilage. **E** Effects of *hoxc12* induction through amputation stress on cell proliferation in the blastema. Statistical test (Student's *t*-test, two-sided): *$p = 1.24 \times 10^{-3}$. Box-plot elements: center line, median; box limits, upper and lower quartiles; whiskers, max/min. All data were obtained from biologically independent samples: $n = 10$ (Ctrl); $n = 10$ (Tg). **F** The positions of tissue sections where immunohistochemistry experiments were performed. **G** The distributions of regenerate nerves (stained with acetylated tubulin) within the mesenchyme (in the distal section in (**F**)). Right: *hoxc12* transgenic *X. laevis*; left: control. The bottom panels are magnified views of the top panels. Epi: epithelium; Mes: mesenchyme; Cart: cartilage. **H** Quantification of regenerated nerves based on the occupancy ratio within the mesenchyme. Statistical test (Student's *t*-test, two-sided): *$p = 3.52 \times 10^{-3}$; **$p = 1.74 \times 10^{-3}$. Source data are provided as a Source Data file.

Tg individuals was higher than in normal spikes (Fig. 4E). We also assessed the histological status of regenerated structures by comparing the quantities of nerves and muscles between Tg and control individuals (Fig. 4F–H). Muscles did not regenerate within the Tg branches like the control spikes (Supplementary Fig. 12). In contrast, the quantity of nerves significantly increased in Tg branches. Specifically, thicker bundles of nerves were observed in the distal regions of the Tg branches (Fig. 4G and Supplementary Fig. 13), and the proportion of nerves occupying the mesenchymal tissue (quantified by the area ratio in a transverse section) was also significantly higher (Fig. 4H).

GFP fluorescence was not directly observable within a Tg blastema without heat shock under a stereo microscope. However, bulk-transcriptome analysis (at 7, 14, and 21 days post-amputation) confirmed that the phenotypic distinction of Tg individuals arose solely by induction of the transgene through amputation stress[28] (see Fig. 5A, B). Note that, because of the high similarity in phenotypes between

*hoxc12*KO and *hoxc13*KO individuals in loss-of-function experiments (Figs. 2 and 3), and between *hoxc12*Tg and *hoxc13*Tg individuals in gain-of-function experiments, we performed comparative transcriptomic analyses only on *hoxc12*Tg individuals. In most *hoxc12*Tg individuals, *gfp* and/or *hoxc12* (transgene) were induced ($n = 15/15, 15/15, 12/15$ at 7, 14, 21 dpa, respectively) (Fig. 5A, B), confirming that the aforementioned phenotypes were caused by the induction of *hoxc12/hoxc13* expression through amputation stress. Since cartilage branching was not observed when much higher expression levels were induced by heat shock, it is likely that an appropriate level of *hoxc12/c13* expression is required to cause branching. In this manner, the phenotypes of regenerated structures in *hoxc12/c13* Tg individuals were clearly different morphologically and histologically from those in normal spikes. However, given that the branched cartilage did not have clear joint-like structures, we could not determine whether it represented regenerated digits or not. Furthermore, the above results indicate that, in order to achieve

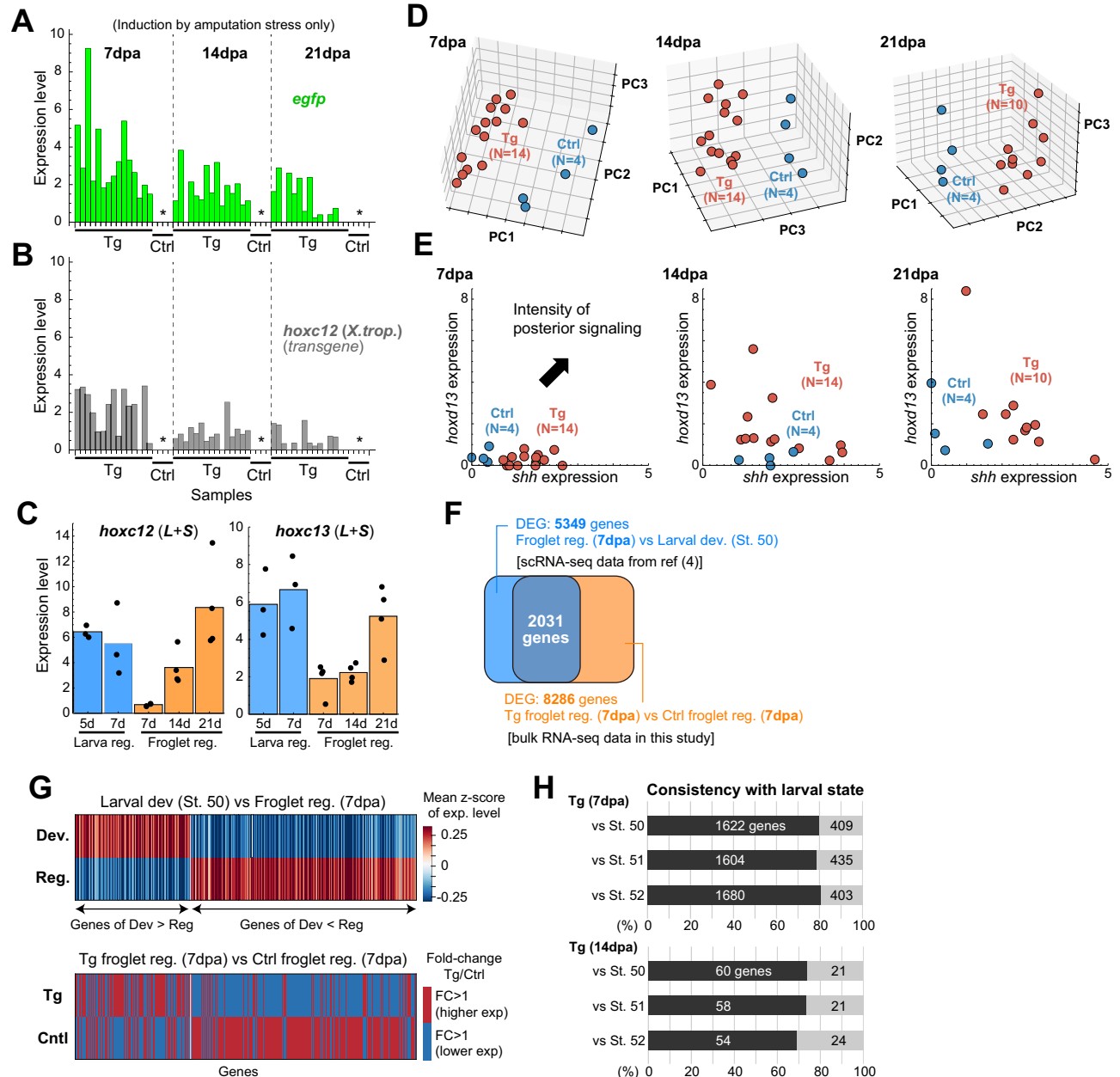

**Fig. 5 | The shift in gene expression profile of Tg-froglet blastema to that of the larval state. A**, **B** Confirmation of the induction of transgene expression **A** *egfp* and **B** *hoxc12* (*X. tropicalis*) solely through amputation stress. *: Not expressed in the control data. **C** Endogenous *hoxc12/hoxc13* expression within the larval/froglet blastema during hindlimb regeneration. For larval regeneration (*Xenopus laevis*), triplicate data sets for regD₁ and regD₂ from Fig. 1 were used, and for froglet regeneration (*Xenopus laevis*), four control data sets shown in Fig. 5A/B (i.e., with no expression of *egfp* and *hoxc12* transgenes) were utilized. Each experiment was independently repeated for larval/ froglet regeneration. Bar: mean. **D** Principal component analysis.

expression space. **F** Extraction of differentially expressed genes (DEGs) between developing limb buds (St. 50) and froglet blastema (7 dpa) based on single-cell transcriptome data from Lin et al. (2021) (referred to as DEG$_{bl-dev}$), and DEGs between Tg/control froglet blastema based on bulk-transcriptome data obtained in this study (referred to as DEG$_{Tg}$). **G** Comparison of expression levels for genes that are common in both DEG$_{bl-dev}$ and DEG$_{Tg}$; (top) developing limb bud vs froglet blastema and (bottom) Tg vs control froglet blastema (see "Methods" section for details). **H** Quantification of the consistency of the shift in gene expression profile in Tg animals toward the expression profile of a developing limb bud. Source data are provided as a Source Data file.

complete regeneration, some rebooting factors other than *hoxc12/c13* are necessary.

### Gene expression shift in Tg-froglet blastema to larval state

Using transcriptome data, we characterized the gene expression profiles of blastema from *hoxc12*Tg individuals. First, we confirmed whether endogenous *hoxc12* and *hoxc13* were expressed within the blastema of control froglets (Fig. 5C; the expression level of each gene was evaluated as the sum of *hoxc12.L* (*hoxc13.L*) and *hoxc12.S* (*hoxc13.S*)

after TPM normalization). Expression levels were low in the earlier stages and increased over time (Fig. 5C). Although a direct evaluation of *hoxc12/c13* expression levels between larval stages and after metamorphosis is difficult due to significant differences in expression profiles of the entire genes, the endogenous expression levels of *hoxc12* and *hoxc13* in the froglet blastema at 21 dpa were comparable to those in the larval blastema at 5/7 dpa. In Tg froglets, the induced expression level of the *hoxc12*-transgene due to amputation stress is higher at 7 dpa, suggesting that increased expression of *hoxc12/c13* at

earlier stages of regeneration (just after initial responses to injury) affects regeneration capacity.

Next, principal component analysis revealed distinct expression patterns of Tg individuals (Fig. 5D). This distinction included changes in the expression of *shh* and *hoxd13*, which are typical posterior factors during development. The expression states of Tg and control individuals were separated in the *shh-hoxd13* expression space (Fig. 5E). However, contrary to the results of our loss-of-function analysis during larval regeneration, when looking at a group of factors known to function during development, there was less of a difference between the expression levels of Tg and control individuals (Supplementary Fig. 14). This is consistent with the observation that complete limb regeneration was not achieved in Tg individuals. Note that, since RNAscope did not work well in froglet tissues, we could not visually confirm the spatial expression profiles of these patterning genes.

Finally, as a crucial indicator that *hoxc12/hoxc13* serves as a regulator in rebooting the developmental program during limb regeneration, we examined whether the induction of *hoxc12/hoxc13* expression shifted gene expression to the larval state. First, using single-cell transcriptome data obtained from Lin et al. (2021) for blastema cells during froglet limb regeneration (at 7 and 14 dpa) and limb bud cells during development (at St. 50, 51, 52), we identified genes differentially expressed between blastema and limb bud cells (approximately 5500 genes, referred to as DEG$_{bl-dev}$) (Fig. 5F (blue) and Supplementary Figs. 15, 16). For each gene within the DEG$_{bl-dev}$, its relative expression level in each cell (among all cells) was computed as a z-score. Subsequently, the average values of the z-scores for all limb bud cells and all blastema cells, respectively, were calculated. When this average value is large, it indicates that the expression level is higher in the focal cell population compared to the other population. As shown in Fig. 5G (top), there are gene groups with higher and lower expression levels in each cell population. Therefore, our objective was to investigate whether induction of *hoxc12/hoxc13* expression in the froglet blastema shifts the gene expression pattern of blastema cells closer to that of the limb bud. To this end, we extracted the differentially expressed genes between Tg and control individuals from our bulk-transcriptome data shown above (Fig. 5F (orange), referred to as DEG$_{Tg}$), and examined the differences in expression levels between them for the common genes in DEG$_{bl-dev}$ and DEG$_{Tg}$ (Fig. 5G (bottom) and Supplementary Figs. 15, 16). At 7 dpa, when the induced expression levels of *gfp* and *hoxc12* (transgene) were relatively high, many genes (~2000) were common to both DEG$_{bl-dev}$ and DEG$_{Tg}$. However, at 14 dpa, when their levels were lower, the number of common genes is reduced (~80 genes, Supplementary Fig. 16) due to the smaller number of DEG$_{Tg}$ (~400 genes). As a significant result, we found that in Tg individuals, the expression state is shifted closer to the larval state (Fig. 5G, H, and Supplementary Figs. 15, 16). In Tg individuals at 7 dpa, approximately 80% of the analyzed genes exhibited the shift in gene expression profile toward the expression profile of a developing limb bud, and at 14 dpa, this was true for approximately 70% of analyzed genes. Taken together, our results support the notion that *hoxc12/hoxc13* functions as a key regulator in rebooting the developmental program during limb regeneration.

## Discussion

In general, the similarity in gene expression patterns between development and regeneration has supported the idea that regeneration recapitulates development[2–4,29]; however, the intrinsic mechanisms for rebooting developmental programs during organ regeneration remain unresolved. Here we demonstrated the roles of *hoxc12* and *hoxc13* as key regulators in rebooting the developmental program during *Xenopus* limb regeneration, although it remains unclear how far upstream these genes operate in the rebooting process. It is noteworthy that there are genes that function only in the morphogenesis phases of regeneration but that have no effect on normal developmental processes. Furthermore, limb regenerative abilities after metamorphosis can be improved, in terms of morphology, nerve formation, and gene expression profiles, solely by modulating the expression level of a single gene that functions during larval regeneration. Another interesting point is the dependence on expression level. While excessive expression of *hoxc12/c13* gene caused the distal blastema to swell, it did not induce the branching that was observed when expression was induced through amputation stress only (Fig. 4B, C). How morphogenesis is affected by the level, duration, and timing of induced expression remains to be elucidated.

Previous studies pointed out that repatterning along the P-D axis is incomplete within the adult *Xenopus* blastema[4,30]. In the blastema of some *hoxc12/c13* transgenic froglets, there were clear morphological differences along the P-D axis, although the genes responsible for P-D patterning during development were not completely upregulated (Fig. 5). In mouse limb development, it has been reported that the exclusion of *Hoxa11* from the *Hoxa13* domain relies on an enhancer that drives antisense transcription at the *Hoxa11* locus after activation by *Hoxa13* and *Hoxd13*[27]. Investigating whether *hoxc12* and *hoxc13* share similar functions could be an interesting avenue for future research. Furthermore, the expression of *shh* and *hoxd13*, which play critical roles in A-P patterning during limb development, were reactivated in the transgenic froglets, which may have caused the A-P expansion of the distal blastema and branching of the distal cartilage. Histologically, nerve regeneration was significantly enhanced, but the regeneration of many tissues contained within the normal limb, including joints and muscles, was not achieved. Simultaneous regulation of multiple genes with high regenerative specificity, as revealed by our transcriptomic analysis (Supplementary Fig. 2), may represent a means for overcoming this problem and is worth exploring in the future.

Finding molecular mechanisms that operate in a regeneration-specific manner will be important for potential applications in animals that lack the ability to regenerate, including humans. For example, the leptin enhancer, which acts specifically during regeneration in zebrafish, is also driven by injury in mice[31]. In mammals, the *hoxc13* gene is expressed in the nail organ during digital tip regeneration as well as during its development[32], and is also involved in maintenance of tissues that undergo repeated production and consumption, such as the hair cycle[33], reminiscent of its association with regenerative ability in other vertebrates. Detailed studies on signaling in these tissues will help us better understand the molecular mechanisms including the *hoxc12/c13* genes in the context of *Xenopus* limb regeneration.

Rebooter genes are assumed to reactivate the developmental program, that is, to simultaneously induce multiple genes related to patterning and growth (*hoxc12/c13* fulfill this role at least during larval regeneration), and thus they are likely to be associated with epigenetic regulation. For example, a limb enhancer region of *shh* is hypomethylated in *Xenopus* tadpoles with higher regenerative capacity, while the sequence is highly methylated in froglets with much lower competence[34]. It was also reported that some Yamanaka factors (Oct3/4, Sox2, and Klf4) and *klf1* are involved in mouse optic nerve regeneration[35] and zebrafish heart regeneration[36], respectively, through epigenetic regulation. Recent single-cell ATAC-seq and SHARE-seq analyses have reported the relationship between chromatin accessibility and *hox* genes[37,38]. These studies suggest the importance of studying regeneration from the perspective of epigenetic regulation through the *hoxc12/c13* genes. Further, it would be interesting to investigate the involvement of *hoxc12/c13* genes in reactive oxygen species (ROS) signaling. Amputation-induced ROS is required for successful *Xenopus* tadpole tail regeneration[39], and ROS is known to regulate a wide variety of signaling pathways including hox families[40]. Identifying rebooter genes that act during various organ regeneration, including *hoxc12/c13* for limb regeneration, and elucidating their common rebooting mechanisms will be an

important step in imparting regenerative capacity to animals that cannot regenerate.

# Methods

## Ethical treatment and manipulation of animals

Experiments on *Xenopus* limb development were conducted at Hiroshima University, Yamagata University, and RIKEN Center for Biosystems Dynamics Research (RIKEN BDR), Japan. Procedures and protocols were approved by the Institutional Animal Care and Use Committee of Hiroshima University. RIKEN and Yamagata University (as well as Japanese domestic law, according to the Act on Welfare and Management of Animals) exempt studies involving amphibians from requiring IRB approval, however, all experiments at RIKEN BDR and Yamagata University were performed in accordance with the principle of 3R (Replacement, Reduction, and Refinement).

Wild-type (WT) *X. tropicalis* adults and tadpoles of the Golden strain were provided by the Amphibian Research Center (Hiroshima University) through the National Bio-Resource Project of the AMED, Japan. WT *X. laevis* adults and tadpoles were purchased from a domestic animal vendor, Watanabe Zoushoku. At the present time, as there is no evidence of gender-based differences in limb regeneration ability, animals were randomly selected regardless of sex. *X. tropicalis* and *X. laevis* embryos/tadpoles/adults were reared at 25–26 °C and 19–23 °C, respectively. The tadpoles were staged according to Nieuwkoop and Faber staging[41]; note that in Fig. 1, we defined the stage for individuals with morphologies between St. 52 and St. 53 as St. 52.5. Tadpoles and froglets were anesthetized in 0.025% ethyl-3-aminobenzoate (MS222, Sigma–Aldrich, A5040), and their limb buds or limbs were excised with spring scissors (Fine Science Tools Inc. 15002-08) or a disposable knife (KAI, 2-5726-28). For in vitro fertilization, after adult male frogs were anesthetized deeply by injecting 500 μL of 1% MS222 and then euthanized, their testes were excised. In vitro fertilization was performed according to previously established protocols[42,43].

## Transcriptome analyses

We performed comparative analyses on bulk-transcriptome data for the following three cases: (Case I) larval hindlimb regeneration vs hindlimb development (Fig. 1), (Case II) larval hindlimb regeneration (*hoxc13*KO vs control, Fig. 3), and (Case III) froglet hindlimb regeneration (*hoxc12*Tg vs control, Fig. 5). Library preparation was performed using the TruSeq Stranded mRNA Library Prep Kit (Illumina) or Illumina Stranded mRNA Prep Ligation Kit (Illumina) with IDT for Illumina RNA UD Indexes (Illumina), and KAPA Real-Time Library Amplification Kit (KAPA Biosystems) for PCR cycle determination. The amount of total RNA used for library preparation was 500 ng for Case I and Case III and 100 ng for Case II. The number of PCR cycles for library amplification was 7 cycles for Case I libraries, 8 cycles for Case III libraries, and 9 cycles for Case II libraries. Sequencing was performed with HiSeq 1500 (Illumina) with the HiSeq SR Rapid Cluster Kit v2 (Illumina) to obtain single-end 80-nt reads or with HiSeq X (Illumina) at Novogene or Azenta to obtain paired-end 150-nt reads. Quality control of the raw sequence reads was performed with FastQC (http://www.bioinformatics.babraham.ac.uk/projects/fastqc/).

In Case I, to compare the gene expression patterns during limb bud regeneration and development (*X. laevis*), bulk RNA sequencing (RNA-seq) was performed for the following regions of interest: the developing limb bud distal or proximal to the prospective knee level at St. 52, 52.5, 53 and 54 (termed $devD_i$ or $devP_i$ in the text, where $i = 1, 2, 3$ or 4 indicating the temporal order), and the regenerating limb bud distal or proximal to the prospective knee level at 5 and 7 dpa (termed $regD_i$ or $regP_i$, where $i = 1, 2$); dpa: days post-amputation. We chose the 5 and 7 dpa time points for the regenerating samples given that the size and morphological appearance of the regenerating blastema ($regD_1$ and $regD_2$) are similar to those of the developing distal limb bud ($devD_1$

and $devD_2$, respectively). Note that, as shown in Fig. 1D, not only the appearance but also the gene expression patterns for $regD_1$ and $regD_2$ are most similar to those for $devD_1$ and $devD_2$, respectively. Total RNA was extracted from the regenerating or developing limb bud using Trizol (Life Technologies, 15596-026) following the manufacturer's protocol. The sequencing was performed in triplicate, where three limb buds or blastemas were pooled for each replicate. After conducting quality control on the raw sequence reads, low-quality reads were further removed using the fastq_quality_filter function of the FASTX Toolkit (v0.0.14) (http://hannonlab.cshl.edu/fastx_toolkit/index.html) as described previously[44]. PhiX, rRNA, and mitochondria-derived sequences were removed by Bowtie (v2.2.2)[45]. The processed reads were mapped to the *X. laevis* genome (J-Strain *X. laevis* 9.1 from Xenbase) by TopHat (v2.0.11)[46]. Expression quantification was performed by Cuffdiff (v2.2.1)[47], and differential expression analysis was performed by edgeR (v3.8.6)[48]. Principal component analysis (PCA) was performed on 10,116 differentially expressed genes (DEGs) using the Python (v3.12.2) scikit-learn library (v1.3)[49] (Fig. 1C). The expression levels of these genes were normalized as Transcripts Per Million (TPM), followed by logarithmic transformation and z-score normalization.

In addition, to identify genes showing regeneration-specific expression changes, we performed three kinds of screening (Fig. 1E, F). In the first screen, 1488 genes differentially expressed during development and regeneration were obtained by comparing the expression level of each gene within each corresponding pair (see the text for the definition of corresponding pair), where differential expression was determined by $p < 0.05$ for each gene. In the second screen, 104 genes showing higher expression levels in the distal region (i.e., blastema) than the proximal region during regeneration were obtained from the genes selected in the first screen, where differential expression was also determined by $p < 0.05$. In the third screen, we identified the 10 genes with the highest regeneration specificity scores (see the text for details on scoring). Gene ontology (GO) annotations of the 104 selected genes were listed using R (v4.2) org.Xl.eg.db package (v3.16.0)[50] and GO.db package (v3.16.0)[51].

In Case II, we compared gene expression profiles in the blastema at 4 dpa among control and *hoxc13*KO tadpoles with normal/severe morphological phenotypes (*X. tropicalis*, quadruplicate, Fig. 3C, D). Each replicate includes three (control, 13KO with normal morphology) or four (13KO with sever phenotype) blastema samples. After conducting quality control on the raw sequence reads, the reads were trimmed by Trim_galore (v0.6.10) (http://www.bioinformatics.babraham.ac.uk/projects/trim_galore/), and the processed reads were mapped to the *X. tropicalis* genome (*X. tropicalis* 10.0 from Xenbase) by STAR (v2.7.9a)[52]. The expression quantification was performed by featureCounts (v2.0.1) [Liao et al., 2014][53]. In Fig. 3C, PCA was performed for typical P-D and A-P patterning genes (listed in Fig. 3D) using the Python (v3.12.2) scikit-learn library (v1.3). In the heatmap in Fig. 3D, the expression level of each patterning gene was shown after z-score normalization among all 12 samples.

In Case III, we compared gene expression profiles in the blastema at 7, 14, and 21 dpa between control and *hoxc12*Tg froglets (*X. laevis*). We prepared 15 replicates for the Tg group and 4 replicates for the control group, where each replicate includes a single blastema sample. After conducting quality control on the raw sequence reads, the reads were trimmed by Trim_galore (v0.6.10), and the processed reads were mapped to the *X. laevis* genome (*X. laevis* 9.2 from Xenbase) by STAR (v2.7.9a). The expression quantification was performed using featureCounts (v2.0.1). As shown in Fig. 5A, B, the amputation stress alone could induce the *hoxc12-transgene*; the transgene expression was detected for 14 samples at 7 and 14 dpa, and 10 samples at 21 dpa. In Fig. 5D, PCA was performed on the expression vectors for all genes using the Python (v3.12.2) scikit-learn library (v4.2), where we used Tg data that showed non-zero counts of transgene expression. In the heatmap shown in Supplementary Fig. 14, the expression levels of typical patterning genes

after z-score normalization were plotted. The data were arranged from left to right in descending order of transgene expression level.

In Fig. 5F−H, differentially expressed genes (DEGs) between froglet-blastemas and developing limb buds were identified using single-cell transcriptome data (gene/cell matrix files from GEO: GSE165901) reported by Lin et al. (2021). Seurat (v4.4.0)[54] was used for the analysis of scRNA-seq data. Initially, we removed low-quality cells and cell multiplets from the dataset, following the methodology described by Lin et al. (2021). As some experiments were conducted using cell pools comprising cells expressing different fluorescent labels, we used marker information to extract targeted stage data. Following normalization with Counts Per Million (CPM), differential expression analysis was carried out using Seurat's FindMarkers function. The number of cells used for the DEG analysis was as follows: 4878 cells at St. 50 (5549 and 5660 cells at St. 51 and 52, respectively) for developmental samples, and 2771 cells at 7 days post-amputation (dpa) (5731 cells at 14 dpa) for regeneration samples. We applied the following criteria for DEGs: (i) FDR (the Benjamini-Hochberg method) <0.01, (ii) fold change >2, and (iii) non-zero expression in 10% or more of cells in each cell population. In Fig. 5G (top), we randomly sampled 1000 cells from both the developmental and regeneration data sets, converted the expression level of each gene in all 2000 cells to a z-score, and calculated the mean z-scores for developmental and regeneration cell populations. For the bulk-transcriptome data obtained at 7 and 14 dpa (Fig. 5G (bottom)), we identified DEGs using DESeq2 (v3.18)[55] after normalization with TPM, applying the criterion of FDR < 0.01.

## Generation of *hoxc12* and *hoxc13* knockout *Xenopus tropicalis* animals using CRISPR-Cas9 genome editing technology

The single guide RNA (sgRNA) was designed as described in Sakane et al.[56] to target the ATG initiation codon of the *hoxc12* or *hoxc13* gene (Fig. 2 and Supplementary Fig. 7). CRISPR-sgRNA injected embryos (crispants) were obtained and genotyped using the Heteroduplex Mobility Assay (HMA) as described in Sakane et al. (2017). Then a sexually mature F0 frog was mated with a wild-type frog, and their offspring were genotyped to screen the F0 frogs with suitable mutation(s) in the germline cells. After *hoxc12*(+/−) mutants were sexually mature (F1 frogs), these female and male mutants were mated to obtain *hoxc12* knockout tadpoles (F2; Fig. 2 and Supplementary Fig. 7). The same experiments were performed to obtain *hoxc13* knockout tadpoles (F2). To obtain *hoxc12* knockout tadpoles more efficiently, F1 frogs were mated with sexually mature *hoxc12* knockout frogs (F2).

F1 and F2 Tadpoles were anesthetized with 0.025% MS222 for photography and tail clipping. For genotyping, a tadpole tail tip (about 5 mm or less) was clipped and lysed in 50 mM NaOH at 96 °C for 15 min following neutralization with Tris-HCl pH 8.0. *hoxc12* and *hoxc13* mutants were identified by PCR analysis of the genomic DNA using the following primers: *hoxc12*, forward primer 5′-GGGTGGGCTTCATGT TTTGG-3′ and reverse primer 5′-CAGATCACAAGCCCTGCTGA-3′; *hoxc13*, forward primer 5′-GATCACGTGTTCCTGGCAGA-3′ and reverse primer 5′-GTTGGAAGAAGACGGGGGAG-3′. The PCR products were prepared for sequencing (FASMAC) using ExoSAP IT™ Express (ThermoFisher, 75001).

## Plasmid constructs, *Xenopus* transgenesis, and heat-shock induction of *hoxc12* or *hoxc13*

Full-length cDNA fragments of *hoxc12* and *hoxc13* were amplified from a cDNA pool of *X. tropicalis* tailbud embryo (St.26) using the following primers:

*hoxc12*, forward primer 5′-ATCGATACCATGGGAGAACATAATCTT CTTAATC-3′ and reverse primer 5′-GTCGACAAAGAATGACAGTGC TTGCTC-3′;

*hoxc13*, forward primer 5′-ATCGATACCATGACGACTTCCCTG ATC-3′ and reverse primer 5′-GTCGAC GGTGTTGTGAAGGTGAGC-3′.

We replaced Mkp3 of the construct, pHS-Mkp3-2A-EGFP/γ-crystallin-tdTomato/IS (kindly gifted by Dr. Hitoshi Yokoyama), with either of the obtained fragments by digesting the construct with ClaI and SalI using the In-fusion cloning system (Takara). pHS-Mkp3-2A-EGFP/γ-crystallin-tdTomato/IS was generated by Dr. Yokoyama through replacing dnTead4 of the pHS-dnTead4-2A-EGFP/γ-crystallin-tdTomato/IS[57] with Mkp3. We prepared pHS-hoxc12-2A-EGFP/γCrystalin-tdTomato or pHS-hoxc13-2A-EGFP/γCrystalin-tdTomato F0 transgenic *X. laevis* by the sperm nuclear transplantation method with oocyte extracts instead of egg extracts[58,59]. We then established stable F1 Tg lines, which were reproduced by crossing sexually mature F0 Tg male frogs with wild-type (WT) females. The transmission of the transgene was determined based on tdTomato expression (as a γ-Crystallin reporter) in the eye. In fact, the transcriptome data indicated that transgenes were expressed in the majority of red-eyed individuals, while none of the black-eyed individuals showed transgene expression (Fig. 5A, B). In Figs. 4D−H and 5, F1 individuals with black eyes were used as controls.

## Localized heat shock

Froglets used were 2 to 2.5 cm in body length, obtained a few days after metamorphosis. After appropriate anesthesia, the hindlimb of pHS-*hoxc12*-2A-EGFP/γCrystalin-tdTomato/IS or pHS-*hoxc13*-2A-EGFP/ γCrystalin-tdTomato/IS F0/F1 transgenic froglets was amputated slightly below the middle of the tibiofibular bone with a disposable knife (KAI, 2-5726-28). Animals were anesthetized for 3 min prior to performing each localized heat shock to the blastema in which the tissue was soaked in hot agarose gel for 30 min; more specifically, we drilled several rectangular holes (4 mm high by 5 mm wide) in a disposable 1000 mL cup (AS ONE Corporation, 1-4621-05), inserted the regenerating hindlimbs into the holes, filled the gap between the limb and cup with adhesive tape, then poured agarose gel at 34 °C to 37 °C into the cup followed by pouring hot water at 50 °C onto the solidifying agarose gel to maintain the temperatures at 34 °C to 37 °C for 30 min (Supplementary Fig. 10). We tested different numbers and durations of heat shock.

## RNAscope assay

Tissues were embedded in OCT compound (Sakura Finetek Japan, 4583) and sectioned at a 10-μm thickness on a cryostat. Fluorescence in situ hybridization was carried out using RNAscope Fluorescent Multiplex Reagent kit (Advanced Cell Diagnostics, #320850) to visualize the spatio-temporal expression of Xl-*fgf8.L* (#500941), Xl-*hoxA11.L* (#850291), Xl-*hoxA13.L* (#850301), Xl-*hoxc12.L* (#900431), Xl-*hoxc13.L* (#894621), Xl-*hoxD13.L* (#891621), Xl-*shh.L* (#850311), Xt-*fgf8* (#899511), Xt-*hoxA11* (#899481), Xt-*hoxA13* (#899491), Xt-*hoxD13* (#899521), and Xt-*shh* (#899501) following the manufacturer's instructions. Briefly, sections were fixed for 15 min at 4 °C with 4% PFA/ PBS then treated with protease IV for 30 min at room temperature. Probes were hybridized for 2 h at 40 °C in a HybEZ II oven. Companion sections were hybridized with negative control probes (#320871) to assure signaling specificity. Sections were counterstained with DAPI (IBC, AR-6501-01) and imaged with a Digital Slide Scanner (Axio Scan Z1, ZEISS) and confocal microscope (LSM880, ZEISS). In Fig. 3A, B, the expression pattern of each gene was classified into three categories (normal, mild, and severe) based on the intensity of expression and the degree of reduction in expression range. Specifically, for *hoxa11*, we classified its expression pattern as normal if its expression level was comparable to that during normal regeneration and if its expression range was the same or wider. We should note that, in some *hoxc12* knockout individuals, *hoxa11* was expressed up to the distal-most region. Such a case was classified as "normal" but is not normal in regard to pattern because *hoxa11* expression is significantly reduced by the exclusive expression with *hoxa13* in the distal-most blastema during normal development and larval regeneration.

## Alcian blue and alizarin red staining

Cartilage and bone staining was performed as described previously[60]. Tissues were fixed with 4% paraformaldehyde overnight at room temperature. To stain the cartilage, samples were dehydrated through a series of ethanol solutions and incubated in Alcian blue solution (0.1 mg/ml Alcian blue 8GX [Sigma, A5268], 80% ethanol, 20% acetic acid) at 37 °C for several hours, until dye deposition was apparent. After rehydration, the samples were treated overnight with 5 mg/ml trypsin (BD Difco, 215240) in 30% saturated $NaB_4O_7$/70% water at room temperature. For bone staining, samples were incubated in 4% alizarin red (Sigma, A5533) saturated with ethanol in 0.5% KOH at room temperature for several hours until dye deposition was apparent. Pigments were bleached in 0.6% $H_2O_2$/1% KO overnight at room temperature.

## Immunohistochemistry

All immunohistochemistry procedures on tissue sections were carried out at room temperature unless otherwise noted. For our cell proliferation assay, tissues were embedded in OCT compound (Sakura Finetek Japan, 4583), sectioned at a 10-μm thickness on a cryostat under −20 °C conditions, and fixed with 4% paraformaldehyde in PBS for 15 min. The sections were then treated 3 times in PBS with 0.5% Triton X-100 for 10 min, blocked with 10% normal goat serum and 3% BSA in PBS for 1 h, followed by incubation with an anti-phosphorylated Histone H3 (PH3) antibody (Millipore, 06-570) diluted at 1:200 in Can Get Signal immunoreaction enhancer solution B (TOYOBO) for 1 h. After washing the sections 3 times in PBS with 0.5% Triton X-100 for 10 min, the sections were incubated with Alexa Fluor 633-conjugated secondary antibodies (Invitrogen, A21050, 1:500) for 1 h. The sections were washed, counterstained with DAPI (IBC, AR-6501-01), and scanned using a digital slide scanner (Axio Scan Z1, ZEISS). In regenerating limb buds of tadpoles, the prospective autopod region was identified as the area expressing *hoxa13*, which was determined using the RNAscope assay on adjacent sections. Within this region, the number of PH3-positive cells per unit area was calculated. For froglet regenerating limbs, the number of PH3-positive cells per area in the blastema was calculated.

For nerve and muscle staining, froglet regenerating limbs were fixed overnight with 4% paraformaldehyde in PBS at 4 °C, then stored in ethanol or Dent's solution (DMSO: Methanol = 1:4) until use. The tissues were gradually immersed in 10%–30% sucrose in PBS at 4 °C before being embedded in OCT compound (Sakura Finetek Japan, 4583) and sectioned by cryostat at 14 μm. Sections were treated 3 times in PBS with 0.01% Triton X-100 for 10 min, then incubated in blocking solution (1% BSA,10% normal goat serum in PBS with 0.01% Triton X-100) for 1 h. The sections were then incubated overnight with primary antibodies (for nerves, anti-acetylated tubulin, Sigma–Aldrich, T7451, 1:1000; for muscles, anti-myosin heavy chain, DSHB, MF 20, 1:20). Sections were diluted in Can Get Signal immunoreaction enhancer solution B (TOYOBO) before being washed 3 times in PBS with 0.01% Triton X-100 for 10 min. Sections were then incubated with Alexa Fluor 633-conjugated secondary antibodies (Invitrogen, A21050, 1:500). After washing, the sections were counterstained with DAPI (IBC, AR-6501-01) and scanned using a digital slide scanner (Axio Scan Z1, ZEISS).

## Hematoxylin-Eosin (HE) staining

We prepared slides for HE staining using the adjacent sections to those utilized for immunostaining of nerves and muscles. Sections were treated with isopropanol for 1 min and dried. HE staining was performed with hematoxylin (Sigma–Aldrich, 51275) for 7 min, bluing buffer (Dako, CS70230-2) for 2 min, and eosin mix solution [Eosin Y (Sigma–Aldrich, HT110216): 0.45 M Tris-Acetic Acid pH 6.0 = 1:9] for 1 min, with each step followed by a wash with water. Slides were scanned by Leica M125 C.

## Real-time PCR

Total RNA was extracted from the regenerating or developing limb bud using Trizol (Life Technologies, 15596-026) and subsequently treated with RNase-free Recombinant DNase I (Takara, 2270A). cDNA was synthesized using the PrimeScript™ RT reagent Kit (Perfect Real Time; Takara, RR037A) following the manufacturer's instructions. Gene expression was analyzed by quantitative PCR (qPCR) using the TaqMan Gene Expression Master Mix (Applied Biosystems, 4369016), the primers and fluorogenic probes for all genes (see Supplementary Table 1) except *odc1* (Assay ID: Xt03690279_g1) were designed using the Custom TaqMan® Assay Design Tool (ThermoFisher Scientific), and 7500 Fast Real-Time PCR System (Applied Biosystems).

## Statistics and reproducibility

In the bulk-transcriptome analysis shown in Figs. 1 and 3D, we used replicate numbers of 3 or 4 to ensure reproducibility of results. In the bulk-transcriptome analysis, we used a higher number of replicates ($n = 15$, for the data of transgenic animals) because not all individuals exhibited the phenotype with cartilage branching and we were unsure if the transgene (*hoxc12*) would be expressed due to limb amputation stress alone in all individuals. We did not employ strict statistical methods to determine the sample size, but given the high reproducibility of the results, we considered these sample sizes to be sufficient. For other experiments, we selected the sample size to ensure data distribution for each group. No data were excluded from the analyses, and animals were selected at random for imaging and gene expression analysis. We did not specifically distinguish between the sexes of the animals. This is because there is no evidence of gender-based differences in regenerative capacity.

## Reporting summary

Further information on research design is available in the Nature Portfolio Reporting Summary linked to this article.

## Data availability

The raw sequence reads generated in this study have been deposited in the NCBI Sequence Read Archive (SRA) under the accession number SRP349043. The single-cell transcriptome data used in this study are available in the GEO database under accession code GSE165901. Source data are provided with this paper.

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

## Acknowledgements

This work was supported by JSPS KAKENHI (20K06664) to A.K.-K. and JST CREST (JPMJCR2025) to Y.M. We would like to thank Prof. Shigehiro Kuraku, Dr. Mitsutaka Kadota, and Dr. Osamu Nishimura for assistance with RNA sequencing and data processing. For providing the pHS-Mkp3-2A-EGFP/γ-crystallin-tdTomato, we thank Dr. Hitoshi Yokoyama. We would like to thank Dr. Haruka Matsubara for construction of the pHS-*hoxc12*-2A-EGFP/γCrystallin-tdTomato and pHS-*hoxc13*-2A-EGFP/γCrystalin-tdTomato plasmids, and also thank Drs. Akihiko Kashiwagi, Keiko Kashiwagi, and the National Bio-Resource Project of *X. tropicalis* in the Amphibian Research Center (Hiroshima University) for providing *X. tropicalis*, Golden strain.

## Author contributions

A.K. and Y.M. designed this study. A.K. primarily conducted experimental work. S.L. and Y.M. performed transcriptome analysis. D.O., K.N., Y.A. and K.K. conducted a portion of the regeneration experiments, as well as genotyping and immunohistochemistry. Y.S. and K.S. generated *hoxc12*/*hoxc13* knockout *X. tropicalis* by genome editing when they were affiliated to Hiroshima University. H.O. created the *hoxc12*/*hoxc13* transgenic *X. laevis*. K.T. provided very helpful comments on the experiments as an expert in limb development and regeneration studies. Y.M. wrote the paper. All authors discussed the results.

## Competing interests

The authors declare no competing interests.
