## [Peer Review File · Nature Communications]

hoxc12/c13 as key regulators for rebooting the developmental program in *Xenopus* limb regenerationREVIEWER COMMENTS

Reviewer #1 (Remarks to the Author):

Kawasumi-Kita et al address a relatively novel regenerative biology concept in their manuscript entitled “*hoxc12/c13* reboots the developmental program in *Xenopus* limb regeneration”. The concept is “re-booting”, which posits the action of post-embryonic and regeneration specific mechanisms that enact embryonic developmental programs to achieve appendage regeneration. Correlational, descriptive and functional data are presented in support of this idea. First, RNA-Seq was performed to screen for differentially expressed genes between developing limb buds and regenerating larval limbs. Criteria were used to identify a short-list of 104 genes that showed higher expression levels in the distal region than proximal region during regeneration, and from this list 2 transcription factors (*HoxC12/HoxC13*) were identified that showed higher expression during regeneration than development. It would be informative to others if an attempt were made to annotate gene names to the 104 short-listed genes as the majority do not have gene acronyms, and with gene names it might be possible to perform an enrichment analysis to determine if there are common ontology themes. At any rate, the two *HoxC* genes were then studied further via in situ hybridization and functional analyses, including CRISPR-Cas9 mediated knockout and transgenic over-expression. In regards to the knockout experiments, *HoxC12/C13* knockout frogs always developed limbs with 5 digits (it would be nice to see an image of this) while proportionally higher numbers of regenerating limbs were characterized by fewer than 5 digits, indicating a disruption of regeneration and thus regeneration specific requirements for these two genes. Arguably this is the most significant result in the manuscript and the initial screening and subsequent analyses showing disruption of patterning genes and partial rescue via transgenic over-expression of *HoxC* genes were included to build a fuller story, which overall is compelling and broadly interesting. Overall, this is a nice body of work

Whether or not *HoxC12/C13* are actually “re-booter” genes is arguable as they apparently function in concert with other genes identified in the initial screen and classical patterning genes that are necessary for regenerative outcomes typical of *Xenopus* larvae. As the authors posit in the discussion, a re-booter mechanism likely acts upstream of *HoxC12/C13* transcription and function and probably involves epigenetic mechanisms that broadly regulate these and other genes. Perhaps this could be delineated more clearly.

Reviewer #2 (Remarks to the Author):

Thank you for the opportunity to review this excellent manuscript from Kawasumi-Kita et al. I think that the readership of Nature Communications will be interested in reading this work, which sets out to identify and test “re-booter” genes for regeneration in the amphibian model organism *Xenopus*. The authors used a sequential analysis to pull out around 100 genes with regeneration-specific expression. *hoxc12* (the only *hox* 12 paralog in *Xenopus*) and *hoxc13* were the top DE genes. These two *hox* genes are expressed at very low levels during limb development from our own observations, and were found to be dispensable for *Xenopus* limb development here. Suemori and Noguchi 2000, PMID: 10753520 had also previously shown that in mice, the entire *hox C* cluster is dispensable for development. Kawasumi-Kita and colleagues first showed that loss of *hoxc12* and -13 expression interfered with the regeneration process that occurs in early developing *Xenopus* limb buds. In *Xenopus*, limb buds can develop normally following loss of the apical ectodermal ridge (AER, FGF8 signalling centre) and developing autopod. In the absence of *hoxc12*, the

autopod region was not re-established, presumably due to the demonstrated failure of clearance of *hoxa11* transcripts from the distal mesenchyme. While this data is convincing, an indication of the level of CRISPR knockdown/editing levels in the F1 tadpoles used would help to support the conclusions.

Next, they created inducible transgenic lines to see if activating these *hox* genes could rescue the normally hypomorphic regeneration of *Xenopus* froglet hindlimbs, using a heat shock construct and polycistronic GFP. Knee level amputations produced branched structures resembling autopods, which demonstrated some consistent re-patterning. Interestingly, these were seen without heat shock, indicating that the amputation itself might be activating this transgene. Heat shock genes have been associated with regeneration in both fish and frogs, so it would be useful to state which heat shock promoter was being used here.

The discussion felt a bit unfocused to me. The loss of exclusion of *hoxa11* from distal regions in the regenerating limb reminded me of the paper on *Hoxa11* and pentadactyly by Kherdjemil et al, 2016 (PMID:27706137). The internal enhancer that drives antisense transcripts of *Hoxa11* is activated distally by *Hox13*, which creates a domain of *hoxa11* exclusion distally in modern tetrapods. I'm not sure if *hoxc12* plays any role in this, but it could explain in part how adding *Hoxc13* improved regeneration. Given the results of this study, I thought that would have made a nice point for discussion and possible future study?

Overall the work is well executed, clearly communicated and interesting. I strongly recommend publication, although I have some minor suggestions that might help to clarify a few details.

1. While the heatmap does provide some support for the similarity groupings, I thought a PCA plot in figure 1 would be useful to not only show this but also to demonstrate how similar the replicate transcriptomes were (since only pooled data is shown but triplicate samples are declared in the methods). There is also a typo in the bottom right of panel 1A – it should read “incomplete”.
2. In Figure 2 B statistics could be applied to this categorical data. Editing levels could be reported for these F1.
3. In the Figure 3D boxplots, are “WT” and “KO *c12* normal” really significantly different as indicated by **, as they do not appear to be differently distributed?
4. In Figure 4, in the absence of HS there was some immediate branching. Was there also GFP expression, indicating the TG was activated by amputation? Was this HSP90 or 70? I think the first question is addressed in supplemental but could also be stated here (or discussed), and I also could not find the origins of the HS promoter. Since different HS proteins have promoters that are differentially responsive to heat, amputation, etc. it would be good to know which is being used here.
5. In methods, please include some detail on the number of samples pooled for each replicate, as I assume these are not from individuals? Also Exo-sap-IT is not a purification method- could say “prepared for sequencing”. I was unsure if the regeneration samples were fore- or hindlimb and what stage they were taken at if not from froglets.
6. *Xenopus*, as it is a genus name, should be in italic, and capitalised.
7. Ethics: the authors state that unconscious frogs were placed in a freezer “for euthanasia”.

Euthanasia means “a good death” and freezing unconscious, but not confirmed dead, animals would be considered unacceptable by ARRIVE or other regional guidelines. I recognise that this is not regulated as extensively in Japan, and I think the statement is just poorly worded, but this does not reflect good ethical practice. Perhaps the wording of this could be revisited?

Caroline W Beck

Reviewer #3 (Remarks to the Author):

The manuscript by Kawasumi-Kita et al describes the identification of genes that are necessary for reinitiating limb developmental programs – “Rebooter genes.” Rebooter genes are genes that are not expressed during limb development but are necessary and sufficient to induce regeneration. While the concept of rebooter genes has been discussed many times in the past, this is an early attempt to identify such genes and their associated phenotypes. To identify such genes authors systematically compared *Xenopus* hind limb development with limb regeneration in larval stages. They identified HoxC12 and HoxC13 as potential rebooter genes. They then used the identified genes to perform loss-of-function and gain-of-function studies in a spike-forming adult stage.

While this is an important concept for the field, it seems to overclaim the results. The following comments must be addressed before its final acceptance.

Major comments:

1. To really make a claim of rebooter genes, comparison of earlier timepoints would be of more interest than later time points of comparing proximal/distal axis. Also, a three-way comparison (limb development, larval regeneration, and adult regeneration) would be more meaningful than a two-way comparison. The presence of unique genes in regenerating blastema as compared to the limb development stage and spike-forming adult stage would rather provide a more meaningful comparison. This criticism stems from three major points.
 - a. To truly claim that HoxC12/13 are rebooter genes, authors must show that fibroblast dedifferentiation is more efficient in presence of HoxC12/13. I encourage authors to look up Lin et al, Dev cell, 2021 paper. It is not necessary that the authors perform single cell seq but even at bulk-seq level – Do you see a shift towards embryonic identity in presence of HoxC12/13.
 - b. The other concern that we have is that the KO and O/X have no phenotypes on early stages of blastema formation but are rather responsible for limb morphogenesis which is a much later stage in limb regeneration. – Suggesting that early stages of blastema formation are not influenced by perturbation of HoxC12/13. In general, when looking for genes that can “reboot” the regeneration process it may be more important to look for the initiators of blastema formation rather than the patterning stage of regeneration.
 - c. Are these genes (HoxC12/13) expressed at similar levels during spike-forming adult blastema stages as compared to the regenerating larval stage? It might be that these genes are not expressed during adult regeneration and that may be why regeneration in adults fails. Overall authors should seriously consider toning down this manuscript, because otherwise it gives a false impression and seems to overclaim the results.
2. Fig.1 G & S3 – What is the pattern of Hoxc12 and Hoxc13 ISH in adult blastema?
3. Figure 3C: Why were hoxc13KO tadpoles not analyzed?
4. Figure 3: The similar patterns & frequencies of abnormalities in individual KO, suggest that a double KO is needed. It could be that the 2 genes substitute for each other’s function.

5. Figure 4 and its associated data are poorly done.

a. In Figure 3, the authors perform systematic experiments for the loss-of-function studies. What happened while doing the same experiments for gain-of-function studies? All the ISH, IHC data need to be produced for gain-of-function studies to make the manuscript more robust.

b. Figure 4B – Lack of phenotype upon heat shock is worrisome and the explanation currently provided is insufficient.

c. Fig. 4C – How do authors know that amputation stress leads to expression of HoxC12 or 13 when there is no GFP or staining for HOXC? ISH can be performed to show induction.

d. Figure 4F, it is unclear what do authors mean by Tg+ -> Is it a gain of function for Hoxc12 or Hoxc13? Or both? If both – please show them individually. If not, both should be done. Also, what is the n? Also, what is -Tg, a non-transgenic animal or Tg animal without heat shock. Please clarify and justify choice of control.

e. Figure 4. for the qPCR data – How many times was the heat-shock given and on what days – particularly because qPCR data are collected at 14 dpa? Do you see GFP expression at 14 dpa (Fig. 4B)?

6. A cross section of O/x limb and control limb with H&E and IHC with neuronal and muscle markers is needed to understand what kinds of changes are observed in the limb of a gain of function animal. Does gain of function lead to more innervation or muscle formation or more connective tissue cells in general? Or is the phenotype restricted to fanning of the distal end of bone?

Minor Comments:

1. Fig. 4D - It might be better to show this graph as a percentage since both C12 and C13 Tg have 33% animals that have phenotype. Can branches among phenotypic animal be quantified?

2. Typos

a. Fig.1a -> incomplete regeneration

b. distalmost -> distal-most

c. Figure 4H: typo-> rebooting instead of rebooting

Positive comment:

1. It is a huge effort to develop germline-transmitted and well-characterized F1/F2 lines. Most previous work in regeneration field produces data with F0 animals – which tends to be less robust.

Replies to reviewers' comments

First, we would like to thank all the reviewers for their very valuable comments and questions. We have listed our responses to the individual comments below.

Replies to Reviewer #1's Comments

Reviewer #1 (Remarks to the Author):

Kawasumi-Kita et al address a relatively novel regenerative biology concept in their manuscript entitled "hoxc12/c13 reboots the developmental program in Xenopus limb regeneration". The concept is "re-booting", which posits the action of post-embryonic and regeneration specific mechanisms that enact embryonic developmental programs to achieve appendage regeneration. Correlational, descriptive and functional data are presented in support of this idea. First, RNA-Seq was performed to screen for differentially expressed genes between developing limb buds and regenerating larval limbs. Criteria were used to identify a short-list of 104 genes that showed higher expression levels in the distal region than proximal region during regeneration, and from this list 2 transcription factors (Hoxc12/Hoxc13) were identified that showed higher expression during regeneration than development. It would be informative to others if an attempt were made to annotate gene names to the 104 short-listed genes as the majority do not have gene acronyms, and with gene names it might be possible to perform an enrichment analysis to determine if there are common ontology themes.

Reply

First, we would like to thank you for your interest in our study. In accordance with your comments, we have added gene ontology annotations to the revised manuscript (Table S1) for the 104 genes selected by transcriptome analysis. We also conducted an enrichment analysis, but only identified generic GO terms, such as involvement in immune and wound repair systems.

At any rate, the two HoxC genes were then studied further via in situ hybridization and functional analyses, including CRISPR-Cas9 mediated knockout and transgenic over-expression. In regards to the knockout experiments, Hoxc12/c13 knockout frogs always developed limbs with 5 digits (it would be nice to see an image of this) while proportionally higher numbers of regenerating limbs were characterized by fewer than 5 digits, indicating a disruption of regeneration and thus regeneration specific requirements for these two genes.

Reply

In the revised manuscript, we have added photographs to demonstrate that limb development is normal

in *hoxc12* or *hoxc13* knockout tadpoles (please see Fig. S4A).

Arguably this is the most significant result in the manuscript and the initial screening and subsequent analyses showing disruption of patterning genes and partial rescue via transgenic over-expression of HoxC genes were included to build a fuller story, which overall is compelling and broadly interesting. Overall, this is a nice body of work. Whether or not Hoxc12/c13 are actually “re-booter” genes is arguable as they apparently function in concert with other genes identified in the initial screen and classical patterning genes that are necessary for regenerative outcomes typical of Xenopus larvae. As the authors posit in the discussion, a re-booter mechanism likely acts upstream of Hoxc12/c13 transcription and function and probably involves epigenetic mechanisms that broadly regulate these and other genes. Perhaps this could be delineated more clearly.

Reply

In the revised manuscript, we have reconsidered the concept of rebooter genes. Specifically, we explicitly stated the idea that if there is a factor responsible for reactivating the developmental program during regeneration, its function and timing could play out in multiple ways. For instance, it is plausible that such a factor could be common to both development and regeneration, or it may function in a regeneration-specific manner. Among these multiple possibilities, in this study, we explored the existence of factors that (i) make a significant contribution to patterning and growth during the morphogenesis phase of limb regeneration, rather than contributing primarily to the initial response to injury, and (ii) have minimal effects on development, but function in a regeneration-specific manner. We found that genes *hoxc12* and *hoxc13* satisfy these conditions.

However, given that the rebooting mechanism has not been fully elucidated (for example, the upstream factors of *hoxc12* / *hoxc13* remain unresolved), it is appropriate that we conclude that *hoxc12* and *hoxc13* are key regulators in rebooting the developmental program during limb regeneration. Consequently, the title has been changed from '***hoxc12/c13* reboots the developmental program in *Xenopus* limb regeneration**' to '***hoxc12/c13* as key regulators for rebooting the developmental program in *Xenopus* limb regeneration**'.

Additionally, please refer to the fourth paragraph of the Introduction for further insights into our motivation and the rebooting process.

Replies to Reviewer #2's Comments

Reviewer #2 (Remarks to the Author):

Thank you for the opportunity to review this excellent manuscript from Kawasumi-Kita et al. I think that the readership of Nature Communications will be interested in reading this work, which sets out to identify and test “re-booter” genes for regeneration in the amphibian model organism Xenopus. The authors used a sequential analysis to pull out around 100 genes with regeneration-specific expression. hoxc12 (the only hox 12 paralog in Xenopus) and hoxc13 were the top DE genes. These two hox genes are expressed at very low levels during limb development from our own observations, and were found to be dispensable for Xenopus limb development here. Suemori and Noguchi 2000, PMID: 10753520 had also previously shown that in mice, the entire hox C cluster is dispensable for development.

Reply

In the revised manuscript, we referenced the paper by Suemori and Noguchi 2000.

Kawasumi-Kita and colleagues first showed that loss of hoxc12 and -13 expression interfered with the regeneration process that occurs in early developing Xenopus limb buds. In Xenopus, limb buds can develop normally following loss of the apical ectodermal ridge (AER, FGF8 signalling centre) and developing autopod. In the absence of hoxc12, the autopod region was not re-established, presumably due to the demonstrated failure of clearance of hoxa11 transcripts from the distal mesenchyme. While this data is convincing, an indication of the level of CRISPR knockdown/editing levels in the F1 tadpoles used would help to support the conclusions.

Reply

In Fig. 2, for *hoxc12*, individuals with a 5-base-pair deletion that included ATG (translation start site) were used in both males and females. For *hoxc13*, individuals with a 12-base-pair deletion/4-base-pair insertion around the ATG site were used in both males and females. In Fig. S7, we examined the effect on phenotype using multiple edited tadpoles for *hoxc13*. For male parents of F1 tadpoles, we used individuals with a 12-base-pair deletion/4-base-pair insertion or 1752-base-pair deletion. For female parents, we used individuals with a 5-base-pair deletion, 40-base-pair deletion, 76-base-pair deletion, and 188-base-pair deletion. As shown in Fig. S7, in genotypes for which enough samples were obtained, morphological phenotypes were observed at frequencies similar to the results shown in Fig. 2B. At least for *hoxc13*, similar phenotypes were observed across multiple edited genotypes, supporting the reproducibility of our results.

In the original/revised manuscript, we stated “The frequency of defects slightly differed depending on the position and amount of the sequence deleted/inserted by genome editing, but, in general, approximately 40-50% of tadpoles (2.5-3 times more than WT) showed abnormalities.”

Next, they created inducible transgenic lines to see if activating these hox genes could rescue the normally hypomorphic regeneration of Xenopus froglet hindlimbs, using a heat shock construct and polycistronic GFP. Knee level amputations produced branched structures resembling autopods, which demonstrated some consistent re-patterning. Interestingly, these were seen without heat shock, indicating that the amputation itself might be activating this transgene. Heat shock genes have been associated with regeneration in both fish and frogs, so it would be useful to state which heat shock promoter was being used here.

Reply

Thank you for this comment. We used hsp70 promoter in our study and have added this information in the revised manuscript.

The discussion felt a bit unfocussed to me. The loss of exclusion of hoxa11 from distal regions in the regenerating limb reminded me of the paper on Hoxa11 and pentadactyly by Kherdjemil et al, 2016 (PMID:27706137). The internal enhancer that drives antisense transcripts of Hoxa11 is activated distally by Hox13, which creates a domain of hoxa11 exclusion distally in modern tetrapods. I'm not sure if hoxc12 plays any role in this, but it could explain in part how adding Hoxc13 improved regeneration. Given the results of this study, I thought that would have made a nice point for discussion and possible future study?

Reply

Thank you very much for this comment. We have added the following sentences in the subsection “**Role of hoxc12/hoxc13 in reactivating patterning and growth**” of the Results:

“The importance of the mutually exclusive expression of hoxa11 and hoxa13 during limb development has also been studied in the context of tetrapod evolution (ref: Kherdjemil et al., 2016).”

And, we also added the following sentences in the Discussion section:

“In mouse limb development, it has been reported that the exclusion of *Hoxa11* from the *Hoxa13* domain relies on an enhancer that drives antisense transcription at the *Hoxa11* locus after activation by HOXA13 and HOXD13. Investigating whether *hoxc12* and *hoxc13* share similar functions could be an interesting avenue for future research.”

Overall the work is well executed, clearly communicated and interesting. I strongly recommend

publication, although I have some minor suggestions that might help to clarify a few details.

1. While the heatmap does provide some support for the similarity groupings, I thought a PCA plot in figure 1 would be useful to not only show this but also to demonstrate how similar the replicate transcriptomes were (since only pooled data is shown but triplicate samples are declared in the methods). There is also a typo in the bottom right of panel 1A – it should read “incomplete”.

Reply

Thank you for this comment. We did a principal component analysis (PCA) and added the result to Fig.1 (please see Fig.1C). For the developmental data, the temporal information (four time points) was neatly arranged and the spatial information (proximal or distal) was well separated in the principal component space. Furthermore, each regeneration dataset (distal or proximal at two time points) was plotted near the corresponding developmental samples that displayed morphological similarities. Additionally, for both developmental and regeneration data, triplicate samples were located close together in the principal component space, indicating high reproducibility of the transcriptome data. Regarding this point, we have added the following sentences to the revised manuscript:

“First, we conducted a principal component analysis and confirmed that each triplicate dataset was located in close proximity within the principal component space, ensuring reproducibility (Fig. 1C). Additionally, in the principal component space, developmental data was neatly arranged along the time axis and proximal-distal (P-D) axis (Fig. 1C). As expected, each regeneration sample (regD₁, regD₂, regP₁, or regP₂) showed a similar expression pattern to that of the developmental sample with similar tissue shape and size (i.e., the pairs regD₁/devD₁, regD₂/devD₂, regP₁/devP₃, and regP₂/devP₄, which are referred to as corresponding pairs for convenience) (Fig. 1A and Fig. S1). This similarity was further validated by quantifying the number of differentially expressed genes across all pairs between development and regeneration samples (Fig. 1D).”

We also thank you for pointing out the typo. We have corrected it.

2. In Figure 2 B statistics could be applied to this categorical data. Editing levels could be reported for these F1.

Reply

As requested, we conducted a statistical analysis for the data shown in Fig. 2B and confirmed that statistically-significant phenotypic abnormalities were observed in individuals with the knockout of *hoxc12/hoxc13*.

3. In the Figure 3D boxplots, are “WT” and “KO c12 normal” really significantly different as indicated by **, as they do not appear to be differently distributed?

Reply

We are sorry for this. As described at the bottom of the Fig. 3 legend, ** indicated “n.s.” in the original manuscript. To avoid this kind of misunderstanding, we have modified it in the revised figure.

4. In Figure 4, in the absence of HS there was some immediate branching. Was there also GFP expression, indicating the TG was activated by amputation? Was this HSP90 or 70? I think the first question is addressed in supplemental but could also be stated here (or discussed), and I also could not find the origins of the HS promoter. Since different HS proteins have promoters that are differentially responsive to heat, amputation, etc. it would be good to know which is being used here.

Reply

Thank you for this important comment. In the revised manuscript, we conducted additional bulk-transcriptome analysis for the blastema (at 7, 14, 21 dpa) of transgenic froglets without HS. As shown in Figs. 5A-B of the revised manuscript, the expression of transgenes (*egfp* and *hoxc12*) is induced solely by amputation stress. In this study, we used HSP70. We have made the following modifications to address the related point:

“we created transgenic (Tg) *X. laevis* with heat-shock-inducible *hoxc12/c13* linked to a reporter GFP via the 2A peptide and γ -crystallin-tdTomato as a reporter for the transmission of transgenes, which is driven by the heat-shock-protein 70 (*hsp70*) promoter (Methods).”

5. In methods, please include some detail on the number of samples pooled for each replicate, as I assume these are not from individuals? Also Exo-sap-IT is not a purification method- could say “prepared for sequencing”. I was unsure if the regeneration samples were fore- or hindlimb and what stage they were taken at if not from froglets.

Reply

Thank you for this comment. In the Methods section, we have specified the number of samples pooled for each replicate and the stage of development (or day post amputation for regeneration) for the three bulk-transcriptome analyses we conducted. Additionally, all analyses were performed using the hindlimb, which has now been clearly stated in the Methods of the revised manuscript.

Regarding Exo-sap-IT, we have modified the Method section as follows:

“The PCR products were prepared for sequencing (FASMAC) using ExoSAP IT™ Express

(ThermoFisher, 75001).”

6. *Xenopus*, as it is a genus name, should be in italic, and capitalised.

Reply

Thank you for pointing this out. We have corrected it.

7. *Ethics: the authors state that unconscious frogs were placed in a freezer “for euthanasia”. Euthanasia means “a good death” and freezing unconscious, but not confirmed dead, animals would be considered unacceptable by ARRIVE or other regional guidelines. I recognise that this is not regulated as extensively in Japan, and I think the statement is just poorly worded, but this does not reflect good ethical practice. Perhaps the wording of this could be revisited?*

Reply

We appreciate this important comment. We have modified the corresponding section in the Methods of the revised manuscript as follows:

“Tadpoles and froglets were anesthetized in 0.025% ethyl-3-aminobenzoate (MS222, Sigma-Aldrich, A5040), and then their limb buds or limbs were excised with spring scissors (Fine Science Tools Inc. 15002-08) or a disposable knife (KAI, 2-5726-28). For in vitro fertilization, after adult male frogs were anesthetized deeply by injecting 500 µL of 1% MS222 and then euthanized, their testes were excised. In vitro fertilization was performed according to previously established protocols (Early Development of *Xenopus laevis*: A laboratory manual, 2000; Viso and Khokha in *Xenopus* protocols: post-genomic approach, 2012).”

Reply to reviewer #3's comments

Reviewer #3 (Remarks to the Author):

The manuscript by Kawasumi-Kita et al describes the identification of genes that are necessary for reinitiating limb developmental programs – “Rebooter genes.” Rebooter genes are genes that are not expressed during limb development but are necessary and sufficient to induce regeneration. While the concept of rebooter genes has been discussed many times in the past, this is an early attempt to identify such genes and their associated phenotypes. To identify such genes authors systematically compared Xenopus hind limb development with limb regeneration in larval stages. They identified Hoxc12 and Hoxc13 as potential rebooter genes. They then used the identified genes to perform loss-of-function and gain-of-function studies in a spike-forming adult stage. While this is an important concept for the field, it seems to overclaim the results. The following comments must be addressed before its final acceptance.

Reply

Thank you for your valuable feedback. As outlined below, in the revised manuscript, we have reconsidered the concept of rebooter genes. Specifically, we explicitly stated our idea that if there is a factor responsible for reactivating the developmental program during regeneration, its function and timing could play out in multiple ways. For instance, it is plausible that such a factor could be common to both development and regeneration, or it may function in a regeneration-specific manner. Among these multiple possibilities, in this study, we explored the existence of factors that (i) make a significant contribution to patterning and growth during the morphogenesis phase of limb regeneration, rather than contributing primarily to the initial response to injury, and (ii) have minimal effects on development but function in a regeneration-specific manner. We found that genes *hoxc12* and *hoxc13* satisfy these conditions.

However, given that the rebooting mechanism has not been fully elucidated (for example, the upstream factors of *hoxc12* / *hoxc13* remain unresolved), it is appropriate that we conclude that *hoxc12* and *hoxc13* are key regulators in rebooting the developmental program during limb regeneration. Consequently, the title has been changed from '***hoxc12/c13* reboots the developmental program in *Xenopus* limb regeneration**' to '***hoxc12/c13* as key regulators for rebooting the developmental program in *Xenopus* limb regeneration**'.

Major comments:

1. *To really make a claim of rebooter genes, comparison of earlier timepoints would be of more interest*

than later time points of comparing proximal/distal axis. Also, a three-way comparison (limb development, larval regeneration, and adult regeneration) would be more meaningful than a two-way comparison. The presence of unique genes in regenerating blastema as compared to the limb development stage and spike-forming adult stage would rather provide a more meaningful comparison. This criticism stems from three major points.

a. To truly claim that *Hoxc12/13* are rebooster genes, authors must show that fibroblast dedifferentiation is more efficient in presence of *Hoxc12/13*. I encourage authors to look up Lin et al, *Dev cell*, 2021 paper. It is not necessary that the authors perform single cell seq but even at bulk-seq level – Do you see a shift towards embryonic identity in presence of *Hoxc12/13*.

Reply

Thank you very much for this valuable comment. For the revised manuscript, we performed bulk-transcriptome analysis on the blastema of *hoxc12*-transgenic froglets. As a crucial indicator that *hoxc12/hoxc13* serves as a regulator in rebooting the developmental program during limb regeneration, we examined whether the induction of *hoxc12/hoxc13* expression shifted gene expression to the larval state. First, using single-cell transcriptome data obtained from Lin et al. (2021) for blastema cells during froglet limb regeneration (at 7 and 14 dpa) and limb bud cells during development (at St. 50, 51, 52), we identified differentially expressed genes (DEGs) between them; that is, the expression differences of them gives the “direction to the larval state”. We extracted the DEGs in the blastema between Tg and control froglets from our new bulk-transcriptome data, and demonstrated that in Tg individuals, the expression state is shifted closer to the larval state. In Tg individuals at 7 dpa (14 dpa), approximately 80% (70%) of the analyzed genes exhibited the shift in gene expression profile toward the expression profile of a developing limb bud. These results strongly support the notion that *hoxc12/hoxc13* functions as a key regulator in rebooting the developmental program during limb regeneration. It should be noted that, because of the high similarity in phenotypes between *hoxc12*KO and *hoxc13*KO individuals in loss-of-function experiments (Figs. 2 and 3), and between *hoxc12*Tg and *hoxc13*Tg individuals in gain-of-function experiments (Fig. 4), we performed comparative transcriptomic analyses only on *hoxc12*Tg individuals.

Please see the third paragraph of the subsection “**The shift of gene expression profile of Tg-froglet blastema to that in the larval state**” and Fig. 5 for more detailed information.

b. The other concern that we have is that the KO and O/X have no phenotypes on early stages of blastema formation but are rather responsible for limb morphogenesis which is a much later stage in limb regeneration. – Suggesting that early stages of blastema formation are not influenced by

perturbation of Hoxc12/13. In general, when looking for genes that can “reboot” the regeneration process it may be more important to look for the initiators of blastema formation rather than the patterning stage of regeneration.

Reply

Thank you for sharing this perspective. The first thing I would like to confirm is that we have explored rebooting factors that reactivate the developmental program during the morphogenesis phase. In this sense, we think that these factors do not necessarily correspond to what you call 'initiators,' which would likely involve factors that regulate the initial response during regeneration. If "rebooter genes" exist, their function and timing could play out in multiple ways. For instance, rebooter genes may directly regulate dedifferentiation during the very early phase of regeneration, just after injury. Alternatively, they may reactivate developmental programs and promote morphogenesis, including axial patterning and growth dynamics similar to those observed during development, after dedifferentiation and initial blastema formation. The latter case may involve molecules that promote "redifferentiation". Compared to dedifferentiation processes, there is particularly little information regarding the self-organization processes of limb morphology during regeneration. Additionally, in *Xenopus* froglet/adult limb regeneration, although dedifferentiation seems to be incomplete, both wound healing and early blastema formation still occur. These observations provided us sufficient motivation to explore the key regulators that function during morphogenesis after an initial response to injury, and to perform comparative transcriptomic analysis between the limb development and morphogenesis phases of larval limb regeneration rather than the very early phase of blastema formation.

As a result, we demonstrated that (i) the knockout of a single gene (*hoxc12* or *hoxc13*) concurrently reduced the expression of dozens of typical patterning genes, which have been intensively studied in the context of vertebrate limb development, during larval regeneration (see the new Figure 3, which includes results from bulk transcriptome analysis). Additionally, we demonstrated that (ii) the overexpression of a single gene alone can partially enhance adult regenerative capacity, and the gene expression profile of the Tg-froglet blastema shifts towards that of the larval state. We think that these results form a foundation for considering that genes *hoxc12* and *hoxc13* play a role in regulating the rebooting of a wide molecular process involved in morphogenesis, rather than regulating a specific signaling pathway.

The motivation for our research, mentioned above, is stated in the fourth paragraph of the Introduction in the revised manuscript.

c. Are these genes (Hoxc12/13) expressed at similar levels during spike-forming adult blastema stages as compared to the regenerating larval stage? It might be that these genes are not expressed during adult regeneration and that may be why regeneration in adults fails. Overall authors should seriously consider toning down this manuscript, because otherwise it gives a false impression and seems to overclaim the results.

Using the above bulk-transcriptome data, we investigated the expression of endogenous *hoxc12* and *hoxc13* within the blastema of control froglets (Fig. 5C). The expression levels were low in the earlier stages and increased over time (Fig. 5C). While a direct evaluation of *hoxc12/hoxc13* expression levels between larval stages and post-metamorphosis is challenging due to significant differences in the expression profiles of entire genes, the endogenous expression levels of *hoxc12* and *hoxc13* in the froglet blastema at 21 dpa were comparable to those in the larval blastema at 5/7 dpa. In Tg froglets, the induced expression level of the *hoxc12*-transgene due to amputation stress is higher at 7 dpa, suggesting that increased expression of *hoxc12/hoxc13* in the earlier stages of regeneration, just after initial responses to injury, influences the regeneration capacity.

With the above revisions (for a, b, and c), we believe that our explanations and conclusions in the revised manuscript neither overstate nor mislead readers.

2. *Fig.1 G & S3 – What is the pattern of Hoxc12 and Hoxc13 ISH in adult blastema?*

Reply

Thank you for bringing up this point. We attempted conventional ISH and RNAscope assays on the froglet blastema multiple times. However, unlike the larval blastema, we were unable to detect a clear signal. The reason for this remains unknown. In this revision, we opted for bulk-transcriptome analysis on the blastema of both Tg and control froglets to enhance our understanding of the expression profile.

3. *Figure 3C: Why were hoxc13KO tadpoles not analyzed?*

Reply

Thank you for your comment. In the original manuscript, we did not conduct quantitative PCR analysis for *hoxc13KO* because the expression patterns of marker genes revealed by the RNAscope assay and the morphological phenotypes were very similar between *hoxc12KO* and *hoxc13KO*. In response to your comment, we performed bulk-transcriptome analysis for *hoxc13KO* tadpoles (please see the revised Fig. 3D). Additionally, we investigated the effect of *hoxc13KO* on cell proliferation (the revised Fig. 3E) and confirmed that, like in *hoxc12KO*, there is a decrease in proliferation rate in the

distal region.

4. Figure 3: The similar patterns & frequencies of abnormalities in individual KO, suggest that a double KO is needed. It could be that the 2 genes substitute for each other's function.

Reply

Thank you for your comment. Given that generating the DKO (double knockout) would be time-consuming and challenging, we have decided, in accordance with the Editor's judgment, not to undertake its creation in this study. Investigating the effects of a DKO and the interdependence of the functions of *hoxc12* and *hoxc13* is a matter we would like to consider for future research.

5. Figure 4 and its associated data are poorly done.

a. In Figure 3, the authors perform systematic experiments for the loss-of-function studies. What happened while doing the same experiments for gain-of-function studies? All the ISH, IHC data need to be produced for gain-of-function studies to make the manuscript more robust.

Reply

We agree with this comment. In order to discern the spatial patterns of gene expression for *hoxc12/c13* and the typical marker genes related to limb development, we attempted ISH for each. However, as mentioned above, it was unsuccessful for reasons we are not yet certain about. In this revision, we conducted bulk transcriptome analysis instead, covering three different time points (7, 14, and 21 dpa). In addition, IHC with a neuronal marker showed the enhancement of nerve regeneration in transgenic individuals.

b. Figure 4B – Lack of phenotype upon heat shock is worrisome and the explanation currently provided is insufficient.

Reply

In the experiments involving heat-shock-induction of transgene (*hoxc12*) expression, we did not observe any branching in the cartilage. However, as mentioned in the main text (Figure 4B in the original manuscript), we did observe a bulge at the distal blastema, which was clearly wider in the A-P direction compared to the wild-type spikes. As confirmed through bulk transcriptome analysis (7, 14, and 21 dpa), the expression of the transgene was induced by amputation stress alone (without heat-shock). Furthermore, when induction was performed by amputation stress alone, GFP fluorescence was not observed under a stereo microscope, indicating a lower expression level compared to heat-shock induction. Based on these findings, we thought that the expression level of *hoxc12* is related to

its impact on the regeneration process.

c. Fig. 4C – How do authors know that amputation stress leads to expression of Hoxc12 or 13 when there is no GFP or staining for HOXC? ISH can be performed to show induction.

Reply

As mentioned above, in the revised manuscript, we performed bulk transcriptome analysis for the blastema of transgenic individuals of *hoxc12* at 7, 14, and 21 dpa. The transgenic individuals we created carry the *Xenopus tropicalis hoxc12* as a transgene. As a result, it is possible to distinguish between the induction of transgene expression triggered by amputation stress alone and the expression level of the intrinsic (i.e., *Xenopus laevis*) *hoxc12*. As demonstrated in revised Figs. 5A-B, we have indeed confirmed the induction of transgenes (*egfp* and *hoxc12*) through amputation stress alone.

d. Figure 4F, it is unclear what do authors mean by Tg+ -> Is it a gain of function for Hoxc12 or Hoxc13? Or both? If both – please show them individually. If not, both should be done. Also, what is the n? Also, what is -Tg, a non-transgenic animal or Tg animal without heat shock. Please clarify and justify choice of control.

Reply

In the revised manuscript, we have decided not to use the terms Tg+ and Tg-. Our updated Methods section now indicates that in our gene expression analysis (i.e., bulk-transcriptome analysis conducted for this revision, the results of which are shown in Fig. 5), we used F1 *hoxc12*Tg lines, which were produced by crossing sexually mature F0 *hoxc12*Tg male frogs with wild-type (WT) females. The transmission of the transgene was determined based on tdTomato expression (as a γ -Crystallin reporter) in the eye. The transcriptome data indicate that transgenes were expressed in the majority of red-eyed individuals, while none of the black-eyed individuals showed transgene expression (Figs. 5A-B). In Figs. 4D-4H and 5, F1 individuals with black eyes were used as controls, and Tg denotes individuals carrying the *hoxc12* transgene; the majority of these individuals exhibit induction of the transgene solely through amputation stress.

In the revised experiment, bulk-transcriptome analysis was conducted on many individuals (Tg+: 15 individuals each for 7, 14, and 21 dpa; Tg-: 4 individuals for each timepoint). Therefore, the results from bulk-transcriptome analyses are presented instead of quantitative PCR in the revised manuscript. 'n' represents the number of individuals. It's important to note that in the qPCR and bulk-transcriptome analyses, the induction of *hoxc12* expression was driven by amputation stress alone, and heat-shock induction was not performed.

In the loss-of-function analysis, both *hoxc12*-KO individuals and *hoxc13*-KO individuals exhibited (quantitatively) the same trends in terms of changes in morphology and in marker gene expression patterns and levels. Furthermore, in gain-of-function analysis, the frequency and shape of the 'with cartilage branch' phenotype in *hoxc13* transgenic individuals closely resembled those of *hoxc12* transgenic individuals. Thus, we conducted a detailed bulk-transcriptome analysis solely for *hoxc12*. We believe that this analysis is sufficient to support our main claim.

e. Figure 4. for the qPCR data – How many times was the heat-shock given and on what days – particularly because qPCR data are collected at 14 dpa? Do you see GFP expression at 14 dpa (Fig. 4B)?

Reply

As mentioned above, in the gene expression analysis, the transgene was induced not by heat shock but solely by amputation stress. Furthermore, at 7, 14, and 21 dpa, induction of *egfp* and/or *hoxc12* (transgene) expression was confirmed in many Tg individuals. In the revised manuscript, we decided to discuss gene expression changes on a genome-wide scale based on bulk-transcriptome analysis data.

6. A cross section of O/x limb and control limb with H&E and IHC with neuronal and muscle markers is needed to understand what kinds of changes are observed in the limb of a gain of function animal. Does gain of function lead to more innervation or muscle formation or more connective tissue cells in general? Or is the phenotype restricted to fanning of the distal end of bone?

Reply

Thank you for this comment. In this revision, we performed immunostaining for markers of neurons and muscles. As a result, we found that the quantity of nerves significantly increased in Tg branches (transgenes were induced by amputation stress only). Specifically, thicker bundles of nerves were observed in the distal regions of the Tg branches (Fig. 4G), and the proportion of nerves occupying the mesenchymal tissue (quantified by the area ratio in a transverse section) was also significantly higher (Fig. 4H). In contrast, muscles did not regenerate within either the Tg branches or the control spikes (Fig. S12). We have also shown H&E staining images in Fig. S12.

Minor Comments:

1. Fig. 4D - It might be better to show this graph as a percentage since both c12 and c13 Tg have 33% animals that have phenotype. Can branches among phenotypic animal be quantified?

Reply

We modified Fig. 4D as suggested.

Regarding the shapes of branches, we decided not to quantify them. The simple spike and branched shapes are visually evident as phenotypes, so in Fig 4D, we examined the frequency of the branched phenotype without quantitative measurement. Quantifying shape variations is theoretically possible (e.g., using elliptic Fourier descriptors), but determining their biological significance and how they relate to other factors is challenging. We judged that this measurement is beyond the scope of this study.

2. Typos

a. Fig. 1a -> incomplete regeneration

b. distalmost -> distal-most

c. Figure 4H: typo-> rebooting instead of rebooting

Reply

Thank you for pointing out these typos. We have corrected them.

REVIEWERS' COMMENTS

Reviewer #1 (Remarks to the Author):

I am satisfied with the revised manuscript, it is much improved and the authors should be commended for generating new data.

Reviewer #2 (Remarks to the Author):

Thank you for the opportunity to review the revised version of this paper. The authors have addressed all minor concerns as detailed in the first round of review, and added extra data to support the conclusions.

Reviewer #3 (Remarks to the Author):

The manuscript by Kawasumi-Kita et al describes the identification genes that are necessary for reinitiating limb developmental programs – “Rebooter genes”. The manuscript is significantly improved, and I want to congratulate the authors for putting in these efforts. While the gain of function data is still weak, I support this manuscript for publication. The issues pertaining to GOF can be addressed by better wording. More specific comments are below.

Major comment

Gain of function:

Fig.4/5:

GOF experiments are luck-dependent due to the complexity of biology, it is ok to acknowledge that in a more candid way.

It is important to acknowledge that some other rebooter genes may be necessary to have better patterning (joint/muscle regeneration etc). Although mentioned in 443-445, it should come early enough while discussing fig.4/5. Also, it is important to acknowledge that very high expression through HSP may be inhibiting the regeneration. It is also possible that transient expression of Hoxc12/c13 as compared to sustained expression by multiple heatshock are required for better patterning. In fact, it would be good to test single pulse of heatshock.

Fig.3 and Fig. S8: FGF8 is barely visible, please consider showing an enlarged image.

Fig. 5C: I am not clear if the results are from HoxC12 Tg or from the wild type (laevis or tropicalis) – a better figure legend is needed.

Minor comment

Fig 3B: Please mention what is N/M/S (normal/mild/severe) in figure legend.

Fig.4F,G,H and Fis. S13: please maintain the order. It is easy for reader, if throughout the manuscript order is maintained. First control and then experimental/test.

Line – 422 - (神経は?),???

- Prayag Murawala

Point-by-point response to the reviewer #3's concerns.

Reviewer #3 (Remarks to the Author):

Major comment

Gain of function:

Fig.4/5:

GOF experiments are luck-dependent due to the complexity of biology, it is ok to acknowledge that in a more candid way. It is important to acknowledge that some other rebooter genes may be necessary to have better patterning (joint/muscle regeneration etc). Although mentioned in 443-445, it should come early enough while discussing fig.4/5. Also, it is important to acknowledge that very high expression through HSP may be inhibiting the regeneration. It is also possible that transient expression of Hoxc12/c13 as compared to sustained expression by multiple heatshock are required for better patterning. In fact, it would be good to test single pulse of heatshock.

Reply

According to the comment, we have added the following sentence in the Results section:

“Furthermore, the above results indicate that, in order to achieve complete regeneration, some rebooting factors other than *hoxc12/c13* are necessary.”

Also, we have modified a sentence in the Discussion section as follows.

Original:

“How morphogenesis is affected by the level of induced expression remains to be elucidated.”

Modified:

“How morphogenesis is affected by the level, duration, and timing of induced expression remains to be elucidated.”

Fig.3 and Fig. S8: FGF8 is barely visible, please consider showing an enlarged image.

Reply

We attempted to enlarge the figures as per the comment. Regarding Fig.3, the visibility did not change significantly, so we chose not to include enlarged figure (please refer to the figure on the last page). Regarding Fig. S8, we replaced the images with higher resolution ones. Additionally, we removed the white dashed lines outlining the regenerating limb buds to improve visibility of the epithelial tissue signal.

Fig. 5C: I am not clear if the results are from HoxC12 Tg or from the wild type (laevis or tropicalis)

– a better figure legend is needed.

Reply

We have added the following sentence in the figure legend for Fig. 5C:

“For larval regeneration (*Xenopus laevis*), triplicate data sets for regD₁ and regD₂ from Fig 1 were used, and for froglet regeneration (*Xenopus laevis*), four control data sets shown in Fig 5A/B (i.e., with no expression of *egfp* and *hoxc12* transgenes) were utilized.”

Minor comment

Fig 3B: Please mention what is N/M/S (normal/mild/severe) in figure legend.

Reply

We have added the information.

Fig. 4F,G,H and Fis. S13: please maintain the order. It is easy for reader, if throughout the manuscript order is maintained. First control and then experimental/test.

Reply

We have changed the order.

Line – 422 - (神経は?),???

Reply

We have modified it.

Fig. 3A

Enlarged images of the distal blastema